# Happy or healthy? How members of the public prioritise farm animal health and natural behaviours

**Belinda Vigors**[1]*, **David A. Ewing**[2], **Alistair B. Lawrence**[1,3]

**1** Scotland's Rural College (SRUC), Edinburgh, United Kingdom, **2** Biomathematics and Statistics Scotland (BioSS), The King's Buildings, Edinburgh, United Kingdom, **3** Roslin Institute, University of Edinburgh, Penicuik, United Kingdom

* belinda.vigors@sruc.ac.uk

**Data Availability Statement:** All relevant data are within the manuscript and its Supporting information files.

## Abstract

The importance given to minimising health issues and promoting natural behaviours is a polarising issue within farm animal welfare. It is predominantly thought that members of the public prioritise animals being able to behave naturally over other aspects of farm animal welfare, such as addressing health issues. However, public perspectives may be more multi-dimensional than is generally thought, with the importance given to these different elements of welfare dependent on the situation and state of the animals in question. To examine this, a factorial survey using vignettes, which experimentally manipulated the different levels of health (high health vs. low health) and natural behaviour provision (high behaviour vs. low behaviour), was completed by a sample (n = 810) representative of the UK population (on age, gender, ethnicity). Contrary to the predominant view, this study found animal health had the greatest effect on participants' judgements, explaining more of the variance in their assessments of animal welfare than any other factor. However, findings also indicated that participants considered animal welfare to be most positive when both health issues are minimised and natural behaviours are promoted. Attitudes to natural behaviours also varied more between participants, with females, individuals who do not (regularly) eat meat and those with a greater belief in animal mind giving greater priority to natural behaviours. In the context of public and private welfare standards seeking to meet public expectations, this study provides important insights into how public perspectives of animal welfare are more nuanced than previously thought, influenced by the context of the animal, the aspect of welfare in question and personal characteristics.

## 1. Introduction

Public concern for animal welfare has always played a key role in the development of farm animal welfare initiatives and policy. From prompting the Brambell report [1, 2], to motivating regulation [3] and to influencing the assessment criteria of welfare standards [3–5], the opinions of particular members of the public and the consumption trends of a sub-set of consumers

**Funding:** The authors gratefully acknowledge funding from the Scottish Government's Rural and Environment Science and Analytical Services Division (RESAS) to conduct this research. The funders had no role in study design, data collection and analysis, decision to publish, or preparation of the manuscript.

**Competing interests:** The authors have declared that no competing interests exist.

have impacted how farm animals are raised and produced and will likely continue to do so [6]. What members of the public (MOP) believe is needed to support good welfare on farms is therefore important.

Numerous social science studies have examined MOP perspectives on farm animal welfare [e.g. 1, 7–10], predominantly in Western or developed countries where public concern for animal welfare continues to rise [6, 11, 12]. For the most part, research indicates that, although MOP consider a variety of factors, they differ from farmers and welfare scientists in the priority they give to 'naturalness' [1, 8, 13, 14] (i.e. the ability of farm animals to express natural behaviours). That is, they want farm animals to retain as much 'normality' as possible by having the opportunity to express their natural behaviours and for artificial interferences, more generally, to be minimal. Hence, public concern tends to be focused on factors deemed to be non-normal or unnatural, such as "restriction of movement in gestation stalls, laying hens in cages, painful procedures (e.g. dehorning of dairy calves and castration of piglets) or lack of natural behaviour" [15, p.1].

Critically, however, the precedence given to 'naturalness' by MOP often diverges from the more practical views of livestock farmers, who largely equate the minimisation of health issues with good welfare [3, 8], and welfare scientists, who consider a diverse range of factors such as biological functioning, affective states and naturalness [3, 16]. For example, Miele et al. [17] found that MOP placed greater importance on animals experiencing positive emotions and living in a natural environment (e.g. outdoor as opposed to confined housing) than welfare scientists. When compared with farmers, Vahonacker et al. [18] found that MOP attitudes to welfare were largely similar except when it came to natural behaviours; MOP considered this to be significantly more important than farmers. The importance MOP give to 'naturalness' has also been argued by those within agriculture to exemplify their lack of knowledge and understanding of modern farming practices [15, 17]. As such, there is an emerging perspective of critical and polarising differences in what key animal welfare stakeholders (e.g. farmers, MOP, welfare scientists) conclude is necessary to improve animal welfare [3, 18].

Despite such divergences and the difficulty in both defining [19] and measuring 'naturalness' [20], there is a convergent belief that what MOP want is for farm animals to have greater opportunities for more natural lives by being able to express natural behaviours [1, 13], with access to the outdoors often considered an important component of this [21, 22]. Such public expectations have led to elements of 'naturalness' (e.g. outdoor-access, minimum days spent at pasture) being considered valid enough for inclusion in both private and state welfare assessment schemes across Europe [5, 20], with retailers, assessment schemes and policy-makers increasingly seeking to be seen to adhere to this desire for 'naturalness' [5, 23, 24]. However, such ideals belie the decision-making conflicts that occur in animal welfare, where priorities can change according to the conditions or system an animal is in [25, 26] and public ideals of 'naturalness' are often at odds with conditions the industry argue are required for animal health, bio-security and cost-effective production [8, 27–29].

Importantly, there is an emerging, but somewhat overlooked, body of research which suggests that when presented with conflicting animal welfare concerns and given the opportunity to consider trade-offs in welfare provisions, MOP may alter their priorities (for recent review see [16]). For example, Cardoso et al. [30] found that MOP demonstrated a greater preference for dairy cows to be kept indoors, rather than outdoors, if this reduced the potential for heat stress. In other words, when faced with this conflicting scenario, MOP chose biological functioning (i.e. lack of heat stress) over 'naturalness' (i.e. pasture access). Such findings suggest that MOP preference for 'naturalness' may not be stable but, instead, may vary according to the animal welfare context and situation in question. Nevertheless, the view that MOP give

precedence to 'naturalness' is now pervasive in the literature and, as previously mentioned, impacts welfare initiatives, policies and assurance schemes.

There thus appears to be critical and polarising differences in the views of key stakeholders (e.g. farmers, MOP, welfare scientists) in relation to animal welfare, along with emergent evidence MOP may alter their priorities for farm animal welfare depending on the context in question (e.g. [30]). In light of these factors, there is a need to more comprehensively understand the extent to which MOP prioritise natural behaviours in their assessments of animal welfare compared to other conflicting concerns (e.g. health issues).

This study examines how UK MOP consider the importance of health and natural behaviours for the well-being of farm animals. More specifically, it aims to determine how MOP ratings of welfare attributes (overall well-being, physical health, mental health and productivity) may vary when assigned to experimental conditions that manipulate the minimisation of health issues and the promotion of natural behaviours. To further separate out the effect of MOP characteristics on welfare-related judgements, this study also examines several between-person factors known to influence attitudes to animal welfare, including socio-demographic factors [31], belief in animal mind [32] and social norms or expectations relating to animal welfare [33]. By doing so, this paper contributes to the development of a clearer understanding of whether and when UK MOP value 'naturalness' in their assessments of animal welfare and the extent to which socio-demographic variables and contextual welfare factors influence MOP judgements of animal welfare. Developing such understanding is important in the context of the UK's recent exit from the EU and an increasing move, within animal production, to a 'pull society' "driven by consumers and facilitated by governments and food retail companies" [34, p.2]. Post-Brexit, to ensure agri-food policy in the UK is effectively informed, there is a need to better understand public expectations relating to farm animal welfare; the findings of this study contribute such needed insights.

## 2. Materials and methods

This study seeks to directly examine the importance MOP place on two central, and often conflicting, elements of good farm animal welfare; the minimisation of health issues and the promotion of natural behaviours. It builds on emerging insights of potential contextual effects by examining how MOP judgements of animal welfare may differ when experimentally assigned to single scenarios which vary the minimisation of health issues (i.e. not minimised vs. minimised) and the promotion of natural behaviours (i.e. promoted vs. not promoted). Based upon the findings of extant literature, it is expected that provision of natural behaviours will have a greater bearing on MOP responses, whereby they will consider welfare attributes to be more positive (i.e. rated higher) when natural behavioural expression is promoted.

To examine this, a factorial survey, employing a 2x2 experimental design, was created and hosted using the Survey Monkey platform (see S1 File). It included slider rating scales, multiple-choice and open-ended questions. The survey was administered in February 2020 to 830 participants whose demographic distribution was reflective of the UK population in terms of gender, age and ethnicity, recruited through Prolific (www.prolific.co). This research and survey were approved by Scotland's Rural College Social Science Ethics Committee and approved as part of the Scottish Government's Rural Affairs Food and the Environment Strategic Research programme. The specific design elements of the survey are detailed in the following sections.

### 2.1. Survey design

**2.1.1 Factorial vignettes.** The purpose of a factorial survey is to more closely reflect real-world decision-making by "force[ing] respondents to make judgements based on trade-offs"

[35, p.11]. This is achieved by presenting participants with a description, or a vignette, of a scenario and asking them to make a judgement on the phenomenon of interest based on the information provided in the vignette. In this study, the primary element of the online survey was a vignette describing how animal health and natural behaviours were managed on a livestock farm. Utilising a 2x2 experimental design, vignettes were defined through the manipulation of two factors; health and natural behaviours, and their two levels; health issues minimised / health issues not minimised and natural behaviours promoted / natural behaviours not promoted. There were thus four possible combinations leading to four vignette scenarios. The four vignettes were randomised between participants so that each individual only received one vignette. To enable brevity in reporting, health issues minimised will henceforth be abbreviated to HH (High Health), health issues not minimised to LH (Low Health), natural behaviours promoted to HB (High Behaviour) and natural behaviours not promoted to LB (Low Behaviour).

To describe scenarios which were reflective of 'real-world' conditions, the vignette wording was generated from descriptions of several livestock farmers, on how they manage animal health and natural behaviours, collected during a prior qualitative interview study (see [36, 37]). Specifically, the vignettes were framed in terms of the hypothetical farmer proactively intervening to minimise health issues (or not) or directly seeking to promote natural behaviours (or not), as opposed to animals simply being healthy or being able to express natural behaviours. The four different vignette scenarios, manipulating the different levels of health and natural behaviours, labelled farm one, farm two, farm three and farm four, are presented in Table 1. It is important to note that participants did not see the vignette labels (e.g. Farm 1: high health x low behaviour), only the vignette narratives. In addition, the vignette section included a descriptor of what was meant by the terms 'health' and 'natural behaviours' (see S1 File).

Based on the information presented in the vignette they were assigned to, participants were asked how they would rate (i.e. judge) several attributes relevant to animal welfare; overall

**Table 1. Vignette scenarios.**

| Farm 1: High Health x Low Behaviour | Farm 2: High Health x High Behaviour |
|---|---|
| "I want my animals to be healthy. To me, this means having them stress free, pain free and injury free, whilst also being aware of any health issues that might be arising and dealing with them. | "I want my animals to be healthy. To me, this means having them stress free, pain free and injury free, whilst also being aware of any health issues that might be arising and dealing with them. |
| At the same time, I don't think I need to do anything specific to support natural behavioural expression in my animals" | At the same time, I want my animals to be able to express their natural behaviours. So, I try to make sure that they can go and have a wander around and see their surroundings, they can choose the animals they want to be around, lie down where they want to lie down and eat when they want to eat" |
| **Farm 3: Low Health x Low Behaviour** | **Farm 4: Low Health x High Behaviour** |
| "When it comes to health, I am inclined to let nature take its course. I'd rather let the animal look after itself than intervene. For example, If I see the odd animal with a sore foot, I'll leave it alone and let it heal in its own time. | "When it comes to health, I am inclined to let nature take its course. I'd rather let the animal look after itself than intervene. For example, If I see the odd animal with a sore foot, I'll leave it alone and let it heal in its own time. |
| At the same time, I don't think I need to do anything specific to support natural behavioural expression in my animals" | At the same time, I want my animals to be able to express their natural behaviours. So, I try to make sure that they can go and have a wander around and see their surroundings, they can choose the animals they want to be around, lie down where they want to lie down and eat when they want to eat" |

well-being, physical health, mental health and productivity (on a sliding scale from 0–10, where descriptors were given to indicate 0 as poor, 5 as average and 10 as excellent). Capturing such judgement-based responses to vignette scenarios reveal what factors can causally affect participants' hypothetical behaviour [35].

A further question was included to assess social norms (i.e. social expectations). Following the recommendations of Bicchieri [38] participants were asked to indicate what rating they thought *other* MOP would give for overall well-being.

The vignette section also included two open-ended qualitative questions asking participants to elaborate on: (i) why they gave the overall rating for well-being that they did, and; (ii) what, if anything, they would change about the farm described in the vignette. This was aimed to provide richer insights on participants' views and judgements of the vignettes.

**2.1.2 Overall attitudes to the importance of health and natural behaviours.** To further understand the importance participants give to health and natural behaviours, an additional section, separate to the vignettes, was included to assess overall attitude to health and natural behaviours. All participants were asked to rate how important they considered (i) minimising health issues and (ii) promoting natural behaviours were for overall animal well-being (on a sliding scale from 0–10, where 0 indicated not important at all, 5 of average importance and 10 extremely important). In addition, they were presented with a binary choice question asking them to select which they considered was the *most important* factor for animal well-being—minimising health issues or promoting natural behaviours.

**2.1.3 Participant characteristics.** There is much evidence to suggest that demographic and personal characteristics, such as age, gender, education, level of income, the type of area a person lives (e.g. rural) and belief in animal mind, influence MOP level of concern for animal welfare [6, 39]. To account for such individual characteristics and their potential to impact judgements relating to health and natural behaviours, the survey collected relevant socio-demographic information. This included gender, age, highest level of education, yearly house-hold income, dietary preferences (i.e. consumes meat, flexitarian, vegetarian, vegan, pescatar-ian), type of area currently living (i.e. urban, sub-urban, semi-rural and rural), geographical region of UK and experience of farming (i.e. no experience, grew up on a livestock farm, friends or family who are livestock farmers, visited livestock farms on educational trips, other experience). Information on participant ethnicity was also available as part of the representa-tive sample provided by Prolific (www.prolific.co). In addition, the belief in animal mind (BAM) scale (as described in [32]) was included to determine the extent to which participants considered farm animals as sentient beings [40]. The BAM scale comprised of four questions which assess the extent to which participants believed farm animals; (i) are unaware of what is happening to them (i.e. not conscious); (ii) capable of experiencing feeling and emotion; (iii) able to think to some extent to solve problems and make decisions; and are (iv) like computer programs, responding to urges without awareness of what they are doing [32, 40]. In line with prior applications of the scale [32] the first and last questions are reversed.

## 2.2 Data preparation

In total, 830 individuals completed the survey. The quantitative data was entered into SPSS, version 25 [41] for analysis. Data was checked for normality, multicollinearity and correlation (for the overall sample and at the level of vignette treatment). Multicollinearity issues (assessed by VIF value >10) were noted with some of the categories for gender, education and dietary preferences. Categories with small sample sizes were thus regrouped or removed from the dataset. This resulted in gender being categorised as male and female (one 'in another way' and three 'prefer not to say' responses were dropped), education categorised as secondary,

undergraduate, post-graduate and other (primary was re-grouped with the 'other education' category) and dietary preferences as 'regularly eat meat', 'flexitarian', 'vegetarian', 'vegan' and 'pescatarian' ('prefer not to say' was dropped and 'other' preferences were re-grouped into applicable categories). Following data checking and the removal of incomplete responses the final sample comprised of 810 participants. The four item BAM scale was examined for reliability, with a Cronbach's alpha of 0.643 indicating a moderate level of internal consistency. This is in line with, and slightly higher than, previous applications of this scale (e.g. [32] reported a Cronbach's alpha of 0.62). A summated scale of BAM was thus created to generate a mean for each participant for their overall BAM (i.e. higher means indicated a greater belief that farm animals are sentient beings). Vignette groups were also checked for balance of demographic factors between groups; there was no evidence of differences.

## 2.3 Statistical analysis

**2.3.1 Analysis of the impact of health provision, natural behaviour provision and participant characteristics on the judgement of vignette welfare attributes.**   A high level of correlation (Cronbach's alpha >.7) was noted between the vignette scenario judgement variables (i.e. rating of overall well-being, physical health, mental health, productivity and social norms) for each individual. Consequently, a multivariate linear regression was used to explore the effect of the different levels of the vignette conditions (i.e. health provision: HH/LH and behaviour provision: HH/LB) and of participant characteristics (i.e. ethnicity, gender, age, income, education, dietary preferences, experience of farming, type of area living, region of UK, BAM) on participants' judgements of welfare attributes (e.g. well-being, physical health etc.). The overall performance of the model was measured by the adjusted $\eta^2$ of value of the model when only terms statistically significant at the 5% level were included (adjusted $\eta^2$ gives the proportion of variance explained adjusted by the number of terms in the model). The model was then refitted including all statistically significant and insignificant terms to give a full picture of the effects of the range of participant characteristics recorded. The significance of terms across the combined dependent variables was assessed using Wilks' $\lambda$, which considers each term having been adjusted for inclusion of the others. Individual effects of each independent variable on single dependent variables were assessed based on partial $\eta^2$. The difference between how each participant rated overall well-being and the rating they gave for how they thought other MOP would rate overall well-being (i.e. social norms) was investigated using a paired t-test.

**2.3.2 Analysis of qualitative responses to vignette scenarios.**   The qualitative responses to (i) what influenced ratings of overall well-being, for each vignette scenario (i.e. farm 1, 2, 3 and 4) were analysed separately using a sentiment analysis and thematic coding approach. Firstly, responses were organised according to sentiment; positive, negative or neutral. This involved first using software (available via Survey Monkey), followed by manual checking, to categorise responses according to how participants felt about the vignette scenario, i.e. whether they described it positively or negatively, or gave neutral responses. For example, describing how the farmer 'cares about the animals' would be taken to indicate a positive response, while describing how the farmer 'could have done more' would be categorised as a negative sentiment. Responses within each sentiment category were then further coded according to the theme of the points discussed. This resulted in overarching sentiment themes and several sub-themes capturing the reasons participants gave for their overall well-being ratings. Responses to (ii) what they would change about the farm described in the vignette, were analysed using a content analysis approach, whereby the commonality of particular descriptive words relating to what participants would change, were assessed.

**2.3.3 Analysis of factors impacting attitude to importance of minimising health issues and promoting natural behaviours.** An exact binomial test was used to assess the binary choice task, where participants' selected which factor they considered the most important for animal well-being—'minimising health issues' or 'promoting natural behaviours'. A paired samples t-test was used to examine differences in ratings (on a scale of 0–10) of how important participants felt each of these factors were for animal well-being.

Ratings of the importance given to 'minimising health issues' and 'promoting natural behaviours' were analysed by fitting a cumulative odds ordinal logistic regression to each response in turn. Both models included terms to account for the effect of participant characteristics and the level of health and behaviour provision (previously exposed to in the vignette scenario).

# 3. Results

## 3.1 Sample

Of the final sample (n = 810) 49% were male and 51% female. In terms of ethnicity, the majority of the sample were White (n = 674), followed by Asian (n = 64), then Black (n = 34), Mixed (n = 21) and those who described their Ethnicity as 'Other' (n = 17). The mean age of the sample was 46 (range 18–88). The majority of participants were educated to secondary level (n = 307), followed by those with an undergraduate degree (n = 296) and a smaller number with post-graduate degrees (n = 151). 56 participants classified their education level as 'Other' which included training such as apprenticeships, higher certificates and vocational qualifications (e.g. NVQ's). An annual household income (before tax) of £20,000 to £34,999 was the most common (n = 237). The majority of respondents lived in urban areas (n = 309) and were spread across all regions of the UK, with the majority based in the North West (n = 102), closely followed by London (n = 101). 50 participants preferred not to disclose their geographic region.

Dietary preferences varied slightly across participants, with a large majority stating they regularly eat meat (n = 505), followed by individuals following a flexitarian diet (e.g. eat meat some of the time) (n = 217), then vegetarian (n = 50), pescatarian (e.g. eat fish but not meat) (n = 23) and vegan (n = 15). The majority of participants had no experience of farm animals or livestock farms (n = 402), followed by those who had been to livestock farms on educational visits (n = 189), those who had friends or relatives who were livestock farmers (n = 114), had 'other' experience (e.g. keep backyard hens, live in an area surrounded by farms) (n = 54), had worked on a livestock farm (n = 27) or grew up on a livestock farm (n = 24). The mean rating for BAM was 6.85 (SD = 1.595). Table 2 presents the demographic data of the sample in detail.

## 3.2 Factorial vignette scenarios

The random assignment of the vignette scenarios resulted in 27% (n = 219) of participants receiving farm one (HHxLB), 23% (n = 189) receiving farm two (HHxHB), 24% (n = 194) receiving farm three (LHxLB) and 26% (n = 208) receiving farm four (LHxHB).

**3.2.1 Judgement of welfare attributes between vignette scenarios.** How participants judged animal welfare, in each scenario, was determined by examining how highly they rated the different welfare attributes of overall animal well-being, physical health, mental health and productivity, based on the information provided on health and behaviour provision in their assigned vignette scenario. Results revealed that farm two (HH x HB) received the highest mean ratings, while farm three (LHxLB) received the lowest mean ratings for each welfare attribute (i.e. well-being, physical health, mental health, productivity). For judgements of overall well-being, physical health and productivity, farm one (HHxLB) received the second highest

**Table 2. Demographic data of study participants.**

|  | Number | % |
| --- | ---: | ---: |
| **Ethnicity** |  |  |
| *White* | 674 | 83 |
| *Asian* | 64 | 8 |
| *Black* | 34 | 4 |
| *Mixed* | 21 | 3 |
| *Other* | 17 | 2 |
| **Gender** |  |  |
| *Male* | 395 | 49 |
| *Female* | 415 | 51 |
| **Age** |  |  |
| *18–30* | 190 | 24 |
| *31–39* | 88 | 11 |
| *40–49* | 128 | 16 |
| *50–59* | 154 | 19 |
| *60 and over* | 238 | 29 |
| *Prefer not to say* | 12 | 2 |
| **Education** |  |  |
| *Secondary* | 307 | 38 |
| *Undergraduate Degree* | 296 | 37 |
| *Post-graduate Degree* | 151 | 19 |
| *Other* | 56 | 7 |
| **Household Income** |  |  |
| *Less than £20,000* | 189 | 23 |
| *£20,000 to £34,999* | 237 | 29 |
| *£35,000 to £49,999* | 149 | 18 |
| *£50,000 to £74,999* | 110 | 14 |
| *£75,000 to £99,999* | 52 | 6 |
| *Over £100,000* | 17 | 2 |
| *Prefer not to say* | 56 | 7 |
| **Dietary Preferences** |  |  |
| *Meat is regular part of diet* | 505 | 62 |
| *Flexitarian* | 217 | 27 |
| *Vegetarian* | 50 | 6 |
| *Vegan* | 15 | 2 |
| *Pescatarian* | 23 | 3 |
| **Type of Area Living** |  |  |
| *Urban* | 309 | 38 |
| *Suburban* | 270 | 33 |
| *Sem-Rural* | 161 | 20 |
| *Rural* | 69 | 9 |
| *Prefer not to say* | 1 | 0 |
| **UK Region** |  |  |
| *London* | 101 | 12 |
| *North East* | 81 | 10 |
| *North West* | 102 | 13 |
| *East Midlands* | 59 | 7 |
| *East of England* | 61 | 8 |

*(Continued)*

**Table 2.** (Continued)

| | Number | % |
|---|---|---|
| *South East* | 96 | 12 |
| *South West* | 74 | 9 |
| *West Midlands* | 56 | 7 |
| *Northern Ireland* | 8 | 1 |
| *Wales* | 51 | 6 |
| *Scotland* | 71 | 9 |
| *Prefer not to say* | 50 | 6 |
| **Experience of Farming** | | |
| *No experience* | 402 | 50 |
| *Grew up on a farm* | 24 | 3 |
| *Relatives/friends farmers* | 114 | 14 |
| *Educational visit to farm* | 189 | 23 |
| *Worked on a farm* | 27 | 3 |
| *Other* | 54 | 7 |

*Note*: n = 810, percentages rounded to nearest whole number

mean ratings, while farm four (LHxHB) the second lowest mean ratings. In contrast, for ratings of mental health, farm four (LHxHB) received the second highest mean rating, while farm one (HHxLB) received the second lowest rating. Fig 1 displays the mean judgement ratings for each welfare attribute across the different levels of health (i.e. LH, HH) and behaviour (i.e. LB, HB) provision presented in the vignette scenarios.

Pairwise comparisons of mean differences indicated that the differences in judgement ratings between the different vignette scenarios were statistically significantly different at the 5% level. Table 3 presents the significant mean differences between each vignette scenario and how they rank in terms of highest to lowest mean rating for each judgement variable (i.e. well-being, physical health, mental health, productivity and social norms). Notably, as indicated by both Fig 1 and Table 3, a scenario which indicated a high level of health provision (i.e. farm one; HHxLB) resulted in higher mean judgement ratings of well-being, physical health, productivity and social norms than the vignette scenario that indicated only behaviour provision was high (i.e. farm four; LHxHB). This is particularly notable when comparing (e.g. in Fig 1) a situation where health provision is high but behaviour provision low, to one where behaviour provision is high but health provision low—compared to high behaviour, high health results in higher mean judgement ratings of all the welfare attributes, except for mental health. Similarly, in Table 2, farm four (LHxHB) was rated third highest for all the judgement variables except mental health, where it was rated second highest.

The social norms judgement captures how highly each participant thought other members of the public would rate the overall well-being of the animals in their assigned vignette scenario. As such, a particular point of interest was the difference between how each participant rated overall well-being and the rating they gave for how they thought other MOP would rate overall well-being. A paired samples t-test indicated that participants who received farm one (HHxLB), farm three (LHxLB) and farm four (LHxHB) scenarios believed other MOP would rate well-being higher than them, with significant respective mean differences of .233 (95% CI, .02 to .45, p = .036), .557 (95% CI, .31 to .81, p <.001) and .317 (95% CI, .12 to .52, p = .002). Participants who received farm two (HH x HB) believed that other MOP would give a lower

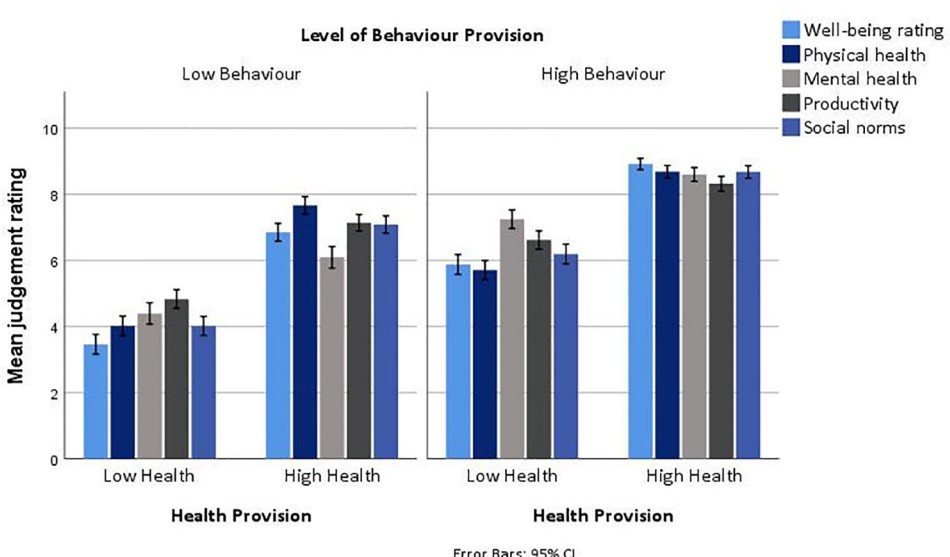

**Fig 1. Mean-rating of well-being, physical health, mental health, productivity and social norms by levels of health and behaviour provision.**

rating for the overall well-being of the animals in the scenario, with a significant mean difference of -.238 (95% CI, -.40 to -.07, p = .005).

**3.2.2 The impact of health provision, natural behaviour provision and participant characteristics on the judgement of welfare attributes.** A multivariate regression, examining how both the different levels of the vignette conditions and participant characteristics impacted judgements of welfare, was fitted. In the model including only statistically significant explanatory variables, the combined predictor variables had the greatest effect on judgements of well-being, $F(47, 809) = 18.73$; $p < .001$; adjusted $\eta^2 = 0.51$ (where adjusted $\eta^2$ gives the proportion of variance explained adjusted by the number of terms in the model). This was followed by physical health, $F(47, 809) = 16.08$; $p < .001$; adjusted $\eta^2 = 0.47$, social norms; $F(47, 809) = 14.40$; $p < .001$; adjusted $\eta^2 = 0.45$, mental health; $F(47, 809) = 10.26$; $p < .001$; adjusted $\eta^2 = 0.35$ and animal productivity, $F(47, 809) = 8.98$; $p < .001$; adjusted $\eta^2 = 0.30$.

In the model including all the explanatory variables (i.e. health and behaviour provision and participant characteristics), the information participants received on health provision (i.e. HH/LH) explained significantly more of the variability of the combined dependent variables than any of other predictor variables, Wilks' $\Lambda = 0.47$; $F(5, 758) = 174.40$; $p < .001$. The information participants received on behaviour provision (i.e. HB/LB) also had a significant effect on the combined dependent variables but this explained less of the variance than health provision; Wilks' $\Lambda = 0.66$; $F(5, 758) = 77.32$; $p < .001$.

Of all the participant characteristic variables included in the multivariate model, only BAM; Wilks' $\Lambda = 0.98$; $F(5, 758) = 3.87$; $p = .002$ and dietary preferences, Wilks' $\Lambda = 0.93$; $F(20, 2515) = 2.90$; $p < .001$, significantly explained some of the variance of the combined dependent variables.

Fig 2 displays the impact of each predictor variable, following adjustment for other variables, on each of the judgement variables for a model with all terms included. Notably, health provision was, again, found to explain more of the variance of each than any other predictor variable. Its strongest significant effect was on judgements of animal physical health; $F(1, 762)$

**Table 3. Pairwise comparison of mean differences between vignette scenarios for judgements of welfare attributes.**

| | Reference Category | | | | Ranking |
|---|---|---|---|---|---|
| | Farm 1 | Farm 2 | Farm 3 | Farm 4 | |
| **Well-being** | | | | | |
| Farm 1 (HHxLB) | 0 | -2.04* | 3.4* | 0.93* | 2nd |
| Farm 2 (HHxHB) | 2.04* | 0 | 5.44* | 2.98* | 1st |
| Farm 3 (LHxLB) | -3.4* | -5.44* | 0 | -2.47* | 4th |
| Farm 4 (LHxHB) | -0.93* | -2.98* | 2.47* | 0 | 3rd |
| **Physical Health** | | | | | |
| Farm 1 (HHxLB) | 0 | -.97* | 3.6* | 1.94* | 2nd |
| Farm 2 (HHxHB) | .97* | 0 | 4.6* | 2.9* | 1st |
| Farm 3 (LHxLB) | -3.6* | -4.6* | 0 | -1.70* | 4th |
| Farm 4 (LHxLHB) | -1.94* | -2.9* | 1.70* | 0 | 3rd |
| **Mental Health** | | | | | |
| Farm 1 (HHxLB) | 0 | -2.51* | 1.72* | -1.2* | 3rd |
| Farm 2 (HHxHB) | 2.51* | 0 | 4.22* | 1.31* | 1st |
| Farm 3 (LHxLB) | -1.72* | -4.22* | 0 | -2.92* | 4th |
| Farm 4 (LHxHB) | 1.2* | -1.31* | 2.92* | 0 | 2nd |
| **Productivity** | | | | | |
| Farm 1 (HHxLB) | 0 | -1.15* | 2.33* | 0.52* | 2nd |
| Farm 2 (HHxHB) | 1.15* | 1 | 3.48* | 1.66* | 1st |
| Farm 3 (LHxLB) | -2.33* | -3.48* | 0 | -1.82* | 4th |
| Farm 4 (LHxHB) | -0.52* | -1.66* | 1.82* | 0 | 3rd |
| **Social Norms** | | | | | |
| Farm 1 (HHxLB) | 0 | -1.53* | 3.16* | 0.95* | 2nd |
| Farm 2 (HHxHB) | 1.53* | 0 | 4.69* | 2.48* | 1st |
| Farm 3 (LHxLB) | -3.16* | -4.69* | 0 | -2.2* | 4th |
| Farm 4 (LHxHB) | -0.95* | -2.48* | 2.2* | 0 | 3rd |

Note:

*shows the mean difference is significant at the.05 level. Adjustment methodology for multiple comparisons: least significant difference.

= 567.74; p<.001; $\eta^2$ = 0.43, followed by well-being; F(1, 762) = 532.12; p<.001; $\eta^2$ = 0.41, social norms; F(1, 762) = 425.56; p<.001; $\eta^2$ = 0.36 and productivity; F(1, 762) = 221.28; p<.001; $\eta^2$ = .23. The only exception to this was judgements of animal mental health, where the information participants received on behaviour provision significantly explained more of the variance, F(1, 762) = 320.94; p<.001; $\eta^2$ = 0.30. Behaviour provision also had a significant (but lower than health provision) effect on judgements of overall well-being; F(1, 762) = 266.97; p<.001; $\eta^2$ = 0.26, social norms; F(1, 762) = 187.06; p<.001; $\eta^2$ = 0.20 and productivity; F(1, 762) = 122.21; p<.001; $\eta^2$ = 0.14. Its lowest effect was on judgements of physical health; F(1, 762) = 94.80; p<.001; $\eta^2$ = 0.11.

Of the overall significant participant characteristic predictors, BAM was found to have a significant effect on judgements of overall well-being; F(1, 762) = 7.22; p = .007; $\eta2$ = .01, and mental health; F(1, 762) = 7.20; p = .007; $\eta2$ = .01. Specifically, BAM was a negative predictor of both, whereby a greater BAM resulted in significantly lower ratings of well-being (b = -.12, SE = .05, p = .007) and mental health (b = -.14, SE = .05, p = .007). This indicates that those with a greater BAM judged the welfare of the animals in each scenario to be less positive than those with a lower BAM. Although dietary preferences explained some of the variance of the

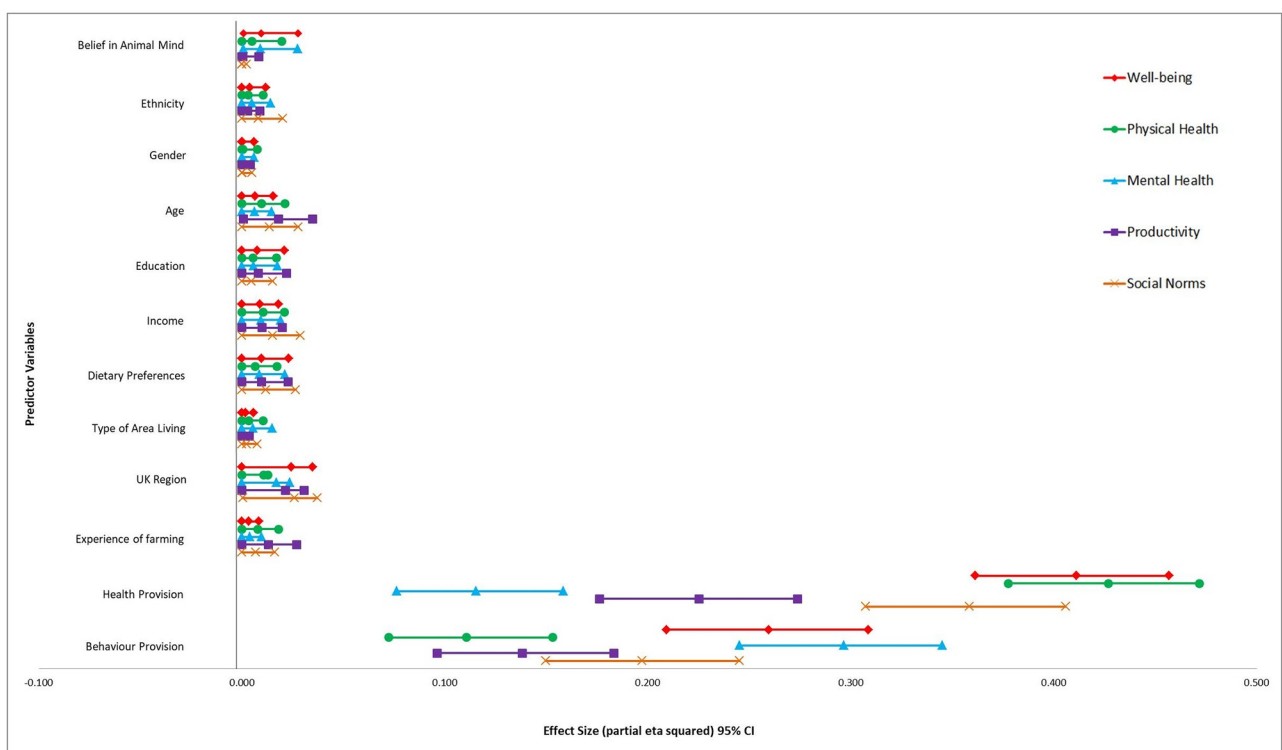

**Fig 2. Impact of participant characteristics and vignette conditions on judgements of well-being, physical health, mental health, productivity and social norms.**

combined dependent variables, it was not found to have a statistically significant effect on any of the dependent variables individually. However, it was notable that vegans and pescatarians gave lower ratings than individuals who regularly eat meat for each judgement variable (i.e. well-being, physical health, mental health, productivity, social norms). Similarly, vegetarians gave lower ratings than meat eaters for all welfare judgement variables except productivity, and flexitarians rated overall well-being and mental health lower than participants who eat meat.

**3.2.3 Qualitative responses to vignette scenarios.** In addition to the quantitative judgement variables, participants were also asked to qualitatively discuss; (i) what influenced their ratings of overall well-being (i.e. why did you give this rating for overall well-being?) and; (ii) what they would change about the farm described in the assigned vignette scenario. With regards to what influences their ratings of overall well-being, Fig 3 presents the sentiment and thematic analysis for each vignette scenario, displaying the percentage of positive, negative or neutral responses within each and the sub-themes relating to the sentiment categories. Neutral responses represented those who responded 'unsure' or 'n/a'.

*3.2.3.1 Farm one*: *health issues minimised x natural behaviours not promoted.* For farm one, sentiment analysis revealed that respondents mostly (58%) perceived this scenario positively, with health issues being minimised presented as the primary reason for positive judgements. Several reasons were given for this, such as minimised health issues serving to indicate that animals were well cared for; "*Animals appear to be treated well and any illness treated accordingly*" and that addressing physical health would contribute positively to overall well-being; "*The animals on the farm are stress free, healthy and illness free, these factors contribute to*

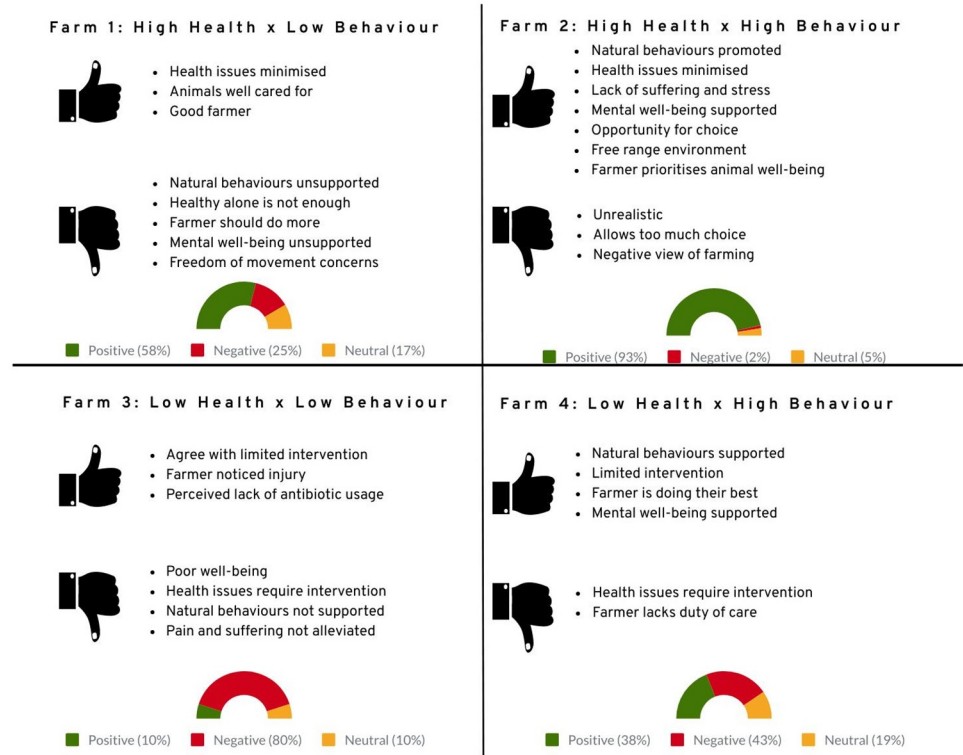

**Fig 3. Sentiment analysis of qualitative responses of reasons given for well-being ratings.** Responses to (ii) what participants would change about the vignette scenario are represented by the word clouds of Fig 4; the size of each word indicates the commonality of particular factors participants would change.

*healthy well-being"*. In addition, numerous positive responses focused on the role and actions of the farmer, judging the farmer in the scenario to be a 'good farmer'; an individual who was caring, put the health of their livestock first and was doing all they could to support the well-being of the animals: *"Because it sounds like the farmer is doing all he can to ensure they are well"*.

However, sentiment analysis did reveal a number of negative responses (25%) to farm one, largely due to natural behaviours not being promoted; *"Animals are not being provided with requirements to express natural behaviours"*. Several further negative themes emerged from this and where interconnected with the latter. Participants described how they believed that minimising health issues alone was not enough for good overall well-being; *"I believe that physical health and welfare should be considered minimal and that….at least some behavioural expression is integral to animal wellbeing"*. Underlying this was a view that overall well-being requires both physical and mental needs, or health and behavioural needs, to be supported; *"The basic needs of the animals are being catered for (i.e. physical health & comfort, food, shelter) but, the 'wellbeing' aspect (i.e. expression of behaviours/mental health) is being ignored"*. A small number of participants also expressed concerns over animals not having freedom of movement; *"The animals are not in obvious distress, but could be unable to move around naturally or express their normal behaviour pattern"*. In addition, many of the negative responses centred on the role of the farmer, describing how they should do more to promote natural behaviours and mental well-being; *"The farmer…could do more to promote their mental well-being"*.

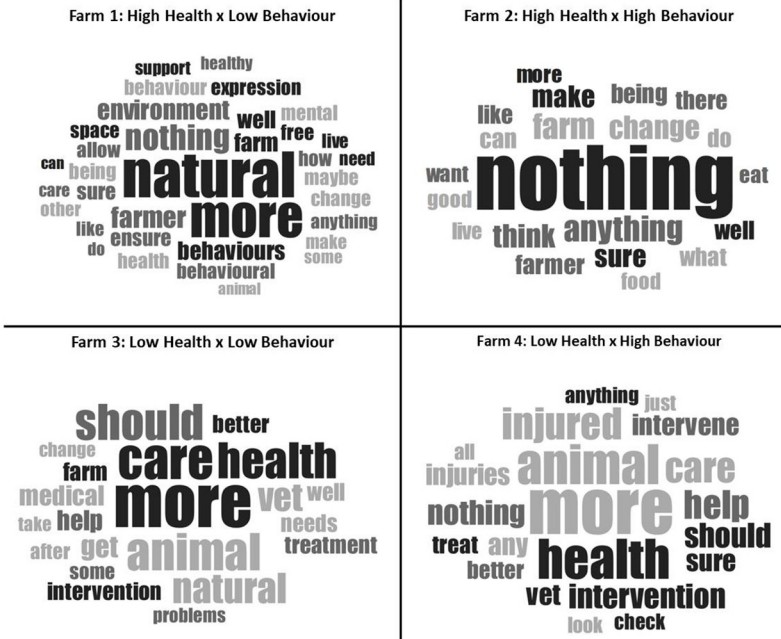

**Fig 4. Wordcloud of qualitative responses to what participants would change about vignette scenarios.**

Notably, several participants expressed a view that farms are 'unnatural' environments and therefore natural behaviours required some direct input from the farmer; "*The farmer is not recognising that captive and domesticated animals often need external input to be able to exhibit natural behaviours in an artificial environment*".

In sum, participants overall had a positive view of this scenario but many highlighted the limitations a lack of promoting natural behaviours may have on the whole well-being of the animal. This view was further illustrated in what participants described they would change about the scenario in farm one. As indicated by Fig 4, doing more to promote natural behaviours was the primary area in which participants wanted changes to be focused.

*3.2.3.2 Farm two*: *health issues minimised x natural behaviours promoted*. Participants' descriptions of what influenced their well-being ratings for farm two were overwhelmingly positive, with sentiment analysis indicating that 93% of responses were positive and only 2% where negative. This was evidenced by the numerous positive adjectives used by respondents' e.g. 'enjoy', 'optimal', 'free', 'happy' and 'encouraged'. The overarching view was that this was an exemplary scenario of high animal welfare that could not be better; "*Within the limits of farming, I can't imagine what more could be done to ensure the animals' well-being*". Several factors in participants' responses indicated why they deduced this. Many referred positively to the fact that both health issues and natural behaviours were taken care of; "*The animals will remain healthy but will still be able to behave naturally, it is almost getting the best of both worlds*", contributing to perceptions that such a scenario would positively support the mental health of the animals; "*Their mental health is also cared for as the animals are able to behave naturally*". Numerous responses also focused on the perceived free range environment and how this would provide animals with 'freedom' and opportunities to exert choice; "*The animals sound as though they have a degree of autonomy and freedom*". The foil of this was numerous references to the lack of animals being confined, 'caged' or space restricted. Similarly, many respondents commented positively on the perceived lack of pain, suffering or stress experienced by

the animals; "*They don't suffer from any injuries or stress needlessly*". A large number of responses also strongly focused on the farmer; praising their actions and commending a perceived proactive approach whereby animal well-being was prioritised and the farmer went 'above and beyond' what was expected; "*The owner of the farm clearly does everything possible to maintain the animals' well-being from both a physical and emotional point of view*". In sum, participants assigned to this scenario viewed it as being particularly positive for an animal's well-being, perceiving it covered all the key elements required for high levels of animal welfare.

The small number of negative responses (2%) were mainly influenced by factors personal to the individual participant. This included having a negative view of livestock farming in general; "*The animals are still captive and raised for food consumption*" and a belief that providing animals with too much freedom or choice was harmful or negative. A small number of participants also questioned the realism of the scenario, feeling it was likely an inaccurate description of most farms.

Responses to what participants would change about this farm further reveal how positively this scenario was viewed. As indicated in Fig 4, the majority of participants said they would change 'nothing' about farm two, further supporting the aforementioned view that this was the most ideal context for supporting the welfare of farmed animals.

*3.2.3.3 Farm three*: *health issues not minimised x natural behaviours not promoted*. Sentiment analysis indicated a strongly negative view (80%) of this scenario. The overarching perception was that animals on this farm would experience poor well-being, mainly due to concerns that health issues were not being treated. The main issue was the lack of intervention; "*I believe that a farmer should intervene to achieve a high standard of health amongst his livestock*" and the pain and suffering that could result from this approach; "*Because the animal is being left potentially in pain and discomfort*". Many participants felt intervention was necessary because they considered farms to be an artificial, as opposed to wholly natural, environment for an animal. As such, intervention to alleviate pain or health issues was considered both possible and necessary; "*Farm animals are not kept in wild conditions or (wild genetic state as selective breeding has occurred) and because they are in a human influenced/created environment with limitations it is the farmer's responsibility to provide medical attention*". Closely related was the view that the farmer, in this scenario, was not fulfilling their duty of care; "*If you run a farm you have a duty of care to look after the animals and should treat any animal who is suffering*" and was therefore providing a bare minimum, or below, level of care; "*it appears to be a minimum of effort being made to look after the animals*". Furthermore, many participants expressed concerns that the farmer's attitude towards minor ailments may lead to more serious health issues being ignored; "*If the farmer does not care about a small thing such as a sore foot, it is likely that he will ignore other more serious conditions*". Interestingly, although natural behaviours were also not supported in this scenario, this was only a minor focus within negative responses with a small number of participants highlighting this as an issue; "*To say he doesn't need to do anything specific to encourage natural behaviour is wrong, farms are not a natural environment for most animals*". Taken together, participants described how these factors led them to consider the farm to be 'below average' and therefore worthy of a low score for overall well-being.

There were a small number (10%) of positive responses. Similar to the negative sentiments, these were also mainly focused on health-related intervention. However, the lack of intervention here was viewed positively and agreed with due to its potential to minimise stress; "*Intervening may cause the animal distress in itself and may not actually be that helpful*". Several respondents also expressed a view that the health issue (a sore foot) was not serious and therefore intervention was not necessary "*For small ailments like the one described (sore foot) this is fine*". Some participants also expresses positive sentiments based simply on the fact that the

farmer had noticed the injury, even if they did not do anything about it; "*The farmer has taken enough interest to at least notice there's a problem*". In addition, a very small number of responses perceived the lack of health intervention would mean minimal use of antibiotics, describing this positively; "*Considering problems caused by over-use of antibiotics/drugs generally—sometimes a 'naturally resolving' attitude for more minor problems is better*". Thus, positive sentiments were, again, primarily focused on health provision rather than natural behaviours.

Responses regarding what participants would change about farm three further indicate that participants primarily focused on the lack of minimising health issues despite the fact that natural behaviours were also not supported. As presented in Fig 4, the majority of participants stated that more health care, veterinary treatment and medical intervention were needed changes on this farm.

*3.2.3.4 Farm four*: *health issues not minimised x natural behaviours promoted*. Sentiment analysis revealed a somewhat mixed response to this scenario, where negative sentiments (43%) only slightly outnumbered positive sentiments (38%). This was evident at the individual level, where most respondents recounted both positive and negative factors influencing the rating they gave for well-being, conveying a sense of having to weigh one against the other to reach a conclusion; "*Allowing the animals freedom of movement is good but not helping them with injuries is not. The number of happy animals would be greater than the number needing help so I gave an overall score of above average*".

Regarding positive sentiments, the majority centred on 'naturalness' where participants perceived this scenario closely mirrored how animals would naturally (i.e. not in farm conditions) live; "*Animals in, predominantly, their natural habitat, eating from the land should be the preferred way for them to live, like they do in the wild*". This included factors such as having freedom of movement; "*I like the animals are allowed freedom of space to naturally explore their surroundings*", being able to exercise some choice or autonomy; "*Allowing animals to be their own decision makers strikes me as more likely for them to 'enjoy' their life*" and being 'free range'; "*The farmer allows the animals to be free instead of keeping them in cages*". This was further connected to a positive perception of natural behaviours being supported in the scenario; "*being able to express natural behaviours is positive for well-being*". In addition, several participants described how they believed the latter two factors (i.e. naturalness and natural behaviours) equated to positive mental well-being; "*This farmer clearly wishes for the animals to be happy (natural behaviours)*". Interestingly, a notable number of respondents also focused on how limited intervention regarding health issues was a positive factor, based on a perception that this was more natural; "*Letting nature take its course is natural and that's how animals have lived for millions of years*". This is a similar attitude to positive sentiments in farm 3, however, in comparison, this sentiment was much more prevalent in responses to this scenario. Positive sentiments were also focused on the role of the farmer, describing them as 'doing their best' and perceived by many to have a caring attitude; "*I can tell the farmer cares for the animals*".

The slight majority of negative sentiments were largely based on a perception that health issues require intervention and the lack of this, in the scenario, was an issue; "*If an animal is injured or ill you need to intervene*". This lack of intervention further led to a negative view of the farmer in this scenario, where several participants equated the lack of intervention to a poor duty of care; "*A livestock farmer has a duty of care for the animal which they are responsible for. Not intervening when seeing injury indicates they are not taking good care*". As such, negative sentiments were much less nuanced than the previously discussed positive responses, being predominantly centred on health issues not being minimised.

In sum, qualitative responses indicated that, compared to all the other scenarios, farm four was the most divisive for participants where conclusions on overall well-being appeared not to be as clear-cut or as easily made as in the other scenarios. Nevertheless, when it came to describing what it was they would change about this farm, a focus on health issues was predominant. As illustrated by Fig 4, the most common response was for there to be more of a focus on health, along with greater intervention to treat injuries.

## 3.3 Attitudes to the importance of minimising health issues and promoting natural behaviours

Beyond the vignette scenarios, which asked participants to make judgements on welfare attributes based on information presented to them, the survey also gathered data on participants' attitude to the importance of health and behaviour provision for animal well-being.

When asked to choose between minimising health issues and promoting natural behaviours, the majority of participants (64%) selected 'minimising health issues' as the most important factor for overall animal well-being (p <.001 under exact binomial test for difference from 50%). When asked to rate (on a scale of 0–10) how important they felt each of these factors were for animal well-being, minimising health issues was rated marginally higher (M = 9, SD = 1.32) than promoting natural behaviours (M = 8.37, SD = 1.67). A paired sample t-test revealed that these importance ratings were statistically significantly higher for minimising health issues than for promoting the expression of natural behaviours by 0.63 (95% CI, 0.52 to 0.74); t(809) = 10.81, p <.001; d = 0.38.

**3.3.1 Factors impacting attitude to importance of minimising health issues and promoting natural behaviours.** To determine the effect of participant characteristics and vignette scenarios on the importance given to minimising health issues and promoting natural behaviours, cumulative odds ordinal logistic regression with proportional odds was run for both outcome variables.

For the importance of minimising health issues, of the participant characteristic predictors, only ethnicity (Wald $x^2$ (4) = 16.29, p = .003) and BAM (Wald $x^2$ (1) = 30.17, p<.001) were found to have a statistically significant effect (having accounted for the inclusion of the other predictors in the model). Of the vignette scenario variables, only health provision had a significant effect, Wald $x^2$ (4) = 75.58, p<.001.

As demonstrated in Table 4, odds ratios indicated that participants of Asian ethnicity were significantly more likely to give a lower rating for the importance of minimising health issues than participants of all other ethnicities. Individuals with a higher BAM were significantly more likely to give a higher rating for the importance of minimising health issues, with an odds ratio of 1.3. In addition, odds ratios indicted that individuals exposed to the high health vignette condition (i.e. health issues minimised) were 3.5 times more likely to give a higher rating for the importance of minimising health issues than an individual exposed to the low health condition (i.e. health issues not minimised).

For the importance given to promoting natural behaviours, gender (Wald $\chi^2$ (1) = 7.85, p = .005), dietary preferences (Wald $\chi^2$ (4) = 20.19, p<.001), experience of farming (Wald $\chi^2$ (5) = 17.46, p = .004) and BAM (Wald $\chi^2$ (1) = 60.96, p<.001) were found to be significant predictors (having accounted for the inclusion of the other predictors in the model). Of the vignette scenario conditions both health provision (Wald $\chi^2$ (1) = 27.38, p<.001) and behaviour provision (Wald $\chi^2$ (1) = 26.00, p<.001) were found to have a significant effect.

Specifically, as indicated in Table 5, odds ratios revealed that females were significantly more likely than males to give a higher rating for the importance of promoting natural behaviours. Odds ratios also indicated that participants who regularly eat meat are significantly less

Table 4. Odds ratios of significant predictors of ratings of the importance of minimising health Issues.

| | OR (95% CI: LL/UL) | | | | |
|---|---|---|---|---|---|
| **Belief in Animal Mind** | **1.3 (1.2/1.4)**\*\*\* | | | | |
| **Health Provision** | *Reference Category* | | | | |
| | *Low Health* | *High Health* | | | |
| *Low Health* | 1 | **.29 (.22/.38)**\*\*\* | | | |
| *High Health* | **3.5 (2.6/4.6)**\*\*\* | 1 | | | |
| **Ethnicity** | *Reference Category* | | | | |
| | *Black* | *Asian* | *Mixed* | *Other* | *White* |
| *Black* | 1 | **3.2 (1.4/7.4)**\*\* | .60 (.20/1.8) | .95 (.30/3.0) | 1.2 (.60/2.5) |
| *Asian* | **.31 (.34/.73)**\*\* | 1 | **.19 (.07/.52)**\*\*\* | **.30 (.10/.98)**\* | **.38 (.22/.66)**\*\*\* |
| *Mixed* | 1.7 (.56/5.1) | **5.4 (1.9/15.0)**\*\*\* | 1 | 1.6 (.43/5.9) | 2.0 (.82/5.1) |
| *Other* | 1.1 (.33/3.4) | **3.4 (1.1/9.9)**\* | .63 (.71/2.3) | 1 | 1.3 (.49/3.4) |
| *White* | .82 (.40.1.7) | **2.6 (1.5/4.5)**\*\*\* | .49 (.20/1.2) | .78 (.29/2.0) | 1 |

Note: OR = Odds Ratio, 95% CI = 95% confidence interval, LL = Lower Level / UL = Upper Level.

\*p<.05,

\*\*p<.01,

\*\*\*p<.001.

likely than flexitarians, vegetarians and vegans to give higher ratings for the importance of natural behaviours. Participants who grew up on a farm were more likely to give lower ratings for the importance of promoting natural behaviours than all the other categories for experience of farming. With regards to information previously conveyed in the vignette scenarios, participants who were exposed to the high health condition were significantly more likely to rate the importance of promoting natural behaviours higher than those exposed to the low health condition. Similarly, individuals who received the high behaviour condition were significantly more likely to give higher ratings for the importance of promoting natural behaviours than those exposed to the low behaviour vignette condition.

## 4. Discussion

This study examined how members of the UK public viewed two of the most central and heavily debated aspects of farm animal welfare—the minimisation of health issues and the promotion of natural behaviours—and the importance given to them under varying conditions. The predominant view, within science and wider society, is that MOP place more importance on natural behaviours, as opposed to health, in their perspectives of animal welfare [6, 39]. The findings of this study challenge this, demonstrating that health issues, overall, have more of an impact on MOP judgements of animal welfare than natural behaviours. Nevertheless, it was also evident that MOP highly value natural behaviours and take into consideration the specific animal welfare context when making related judgements. As such, this study sheds a more detailed light on when and for what reasons health issues and natural behaviours are considered important and the various factors underlying and contributing to the views and judgements of MOP.

Importantly, the findings of this study clearly indicate that what UK MOP want is for farm animals to be *both* healthy and able to engage in natural behaviours. This is evidenced by the HHxHB scenario (farm two) receiving the highest ratings for all of the welfare judgement attributes and the overwhelmingly positive qualitative responses to this scenario. Furthermore,

**Table 5. Odds ratios of significant predictors of ratings of the importance of promoting natural behaviours.**

|  | OR (95% CI: LL/UL) |  |  |  |  |  |
|---|---|---|---|---|---|---|
| **Belief in Animal Mind** | 1.4 (1.3/1.6)*** |  |  |  |  |  |
| **Health Provision** | *Reference Category* |  |  |  |  |  |
|  | *Low Health* | *High Health* |  |  |  |  |
| *Low Health* | 1 | **.50 (.38/.65)*** |  |  |  |  |
| *High Health* | **2.0 (1.5/2.6)*** | 1 |  |  |  |  |
| **Behaviour Provision** | *Reference Category* |  |  |  |  |  |
|  | *Low Behaviour* | *High Behaviour* |  |  |  |  |
| *Low Behaviour* | 1 | **.51 (.39/.66)*** |  |  |  |  |
| *High Behaviour* | **1.97 (1.5/ 2.6)*** | 1 |  |  |  |  |
| **Gender** | *Reference Category* |  |  |  |  |  |
|  | *Males* | *Females* |  |  |  |  |
| *Males* | 1 | **.68 (.51/.89)** |  |  |  |  |
| *Females* | **1.5 (1.1/2.0)** | 1 |  |  |  |  |
| **Dietary Preferences** | *Reference Category* |  |  |  |  |  |
|  | *Flexitarian* | *Vegetarian* | *Vegan* | *Pescatarian* | *Regularly Meat* |  |
| *Flexitarian* | 1 | .70 (.38/1.3) | .38 (.11/1.3) | 1.5 (.66/3.5) | **1.7 (1.2/ 2.3)*** |  |
| *Vegetarian* | 1.4 (.79/2.6) | 1 | .55 (.15/2.0) | 2.1 (.82/5.8) | **2.4 (1.3/4.3)** |  |
| *Vegan* | 2.6 (.78/8.8) | 1.8 (.51/6.6) | 1 | 4.0 (.96/16.4) | **4.3 (1.3/14.4)*** |  |
| *Pescatarian* | .66 (.29/1.5) | .46 (.17/1.2) | .25 (.06/1.0) | 1 | 1.1 (.49/2.5) |  |
| *Regularly Meat* | **.60 (.44/.82)*** | **.42 (.23/.75)** | **.23 (.07/.77)*** | .91 (.41/2.1) | 1 |  |
| **Experience of farming** | *Reference Category* |  |  |  |  |  |
|  | *Grew up on farm* | *Relatives/Friends Farmers* | *Educational visit to farm* | *Have worked on a farm* | *Other* | *No experience of farming* |
| *Grew up on farm* | 1 | **.30 (.13/.68)** | **.20 (.09/.44)*** | **.23 (.08/.65)** | **.317 (.13/ .76)** | **.24 (.11/.51)*** |
| *Relatives/Friends Farmers* | **3.3 (1.5/7.5)** | 1 | .68 (.43/1.1) | .76 (.33/1.8) | 1.1 (.57/2.0) | .80 (.53/1.2) |
| *Educational visit to farm* | **4.9 (2.3/ 10.7)*** | 1.5 (.93/2.3) | 1 | 1.1 (.49/2.6) | 1.6 (.88/2.8) | 1.2 (.85/1.6) |
| *Have worked on a farm* | **4.4 (1.5/12.6)** | 1.3 (.56/3.1) | .89 (.39/2.0) | 1 | 1.4 (.56/3.4) | 1.1 (.47/2.4) |
| *Other* | **3.2 (1.3/7.5)** | .94 (.51/1.7) | .64 (.36/1.1) | .72 (.29/1.8) | 1 | .76 (.44/1.3) |
| *No experience of farming* | **4.2 (1.9/8.9)*** | 1.2 (.82/1.9) | .85 (.61/1.2) | .95 (.44/2.1) | 1.3 (.77/2.3) | 1 |

Note: OR = odds ratio, 95% CI = 95% confidence interval, LL = lower level / UL = upper level.

*p<.05,

**p<.01,

***p<.001.

qualitative responses to scenarios where only health was supported (i.e. farm one) strongly emphasised the need to additionally support natural behaviour. Such findings indicate that MOP consider the well-being of farm animals to be most positive when both health and natural behaviours are supported and no trade-offs have to be made between them. This fits well with the growing 'positive animal welfare' literature, which argues that for animals to live a life worth living, there is a need for both negative factors affecting them to be minimised and positive factors promoted [42–47]. Further similarity with the core arguments of the positive welfare literature [e.g. 43, 46, 47] is evident in the qualitative responses, where participants emphasised that minimising health issues alone was not enough for animals to experience

positive well-being; natural behaviour expression was described as a requirement to support positive experiences (e.g. positive affect, positive mental health and happiness) in farm animals.

Critically, however, results of this study indicate that the UK public appear to place more importance on health issues being minimised. When asked to choose between minimising health issues and promoting natural behaviours, the majority chose minimising health issues. In addition, when asked to rate how important each was for animal well-being, minimising health issues was rated marginally more important. Moreover, when comparing the scenarios where there was some trade-off between health and behaviour provision (i.e. farm one and farm four), situations where health issues were minimised but natural behaviours were not promoted (farm one) were considered better for the majority of welfare-related judgements than situations where natural behaviours were promoted but health issues were not minimised (farm four). Indeed, health provision was found to explain more of the variance for each welfare-related judgement variable (except for mental health) than any other, including those relating to participant characteristics. This arguably indicates that although MOP expressed an attitudinal desire for both health and natural behaviours to be supported (as revealed in qualitative responses), their actual judgement decisions were mostly influenced by the level of health provision. In short, health appeared to *matter* more to participants than natural behaviours when making assessments of animal well-being and welfare.

Such a finding conflicts with the predominant view that 'naturalness' is the most important aspect of welfare for MOP (e.g. [1, 8, 14]). This may, in part, be due to the nature of this study's sample (being representative of UK population demographics on age, gender and ethnicity)—they were what Miele [48, p.1] refers to as "ordinary citizens". Unlike participants in extant research who are often non-representative and, in some cases, are purposively selected for having an *a priori* interest in animal welfare (e.g. [8]), participants in this study were unlikely to have pre-existing knowledge of animal welfare. As individuals with prior knowledge of animal welfare tend to be more concerned about animal welfare ([49] cited in [6]) it is therefore possible that the participants in this study may give less importance to factors such as natural behaviours compared to the samples typical of extant research. Welfare Quality® research has also suggested that MOP tend to believe that animal suffering is largely negated in Europe and, therefore, welfare should be used only to refer to positive factors such as 'outdoor access' and a 'natural life' [48]. Consequently, it could be theorised that many MOP emphasise the importance of natural behaviours because they believe, at a minimum, health issues are already taken care of and, therefore, natural behaviour provision requires attention. The fact that participants in this study had to simultaneously consider health and behaviour provision (due to the factorial survey design) may thus explain why health provision had a greater impact on MOP judgement of welfare; when it was made evident that health issues may or may not be minimised, they gave greater priority to health provision over behaviour provision. As such, MOP may take a more holistic view of animal welfare, giving different weight to health and behaviour [39] particularly when "weighing the concern that animals should lead a natural life, against suffering" [1, p.229].

Results thus support and build on the emergent research suggesting MOP alter their priorities for animal welfare according to the nature of the situation in which an animal is in [16, 30]. Participants' qualitative responses revealed how MOP consider numerous contextual factors, often simultaneously. Mention of factors such as the severity of an injury, freedom of movement/autonomy and opportunities to experience enjoyment or happiness suggest that participants were not only considering whether health or behaviours were supported but also the actual negative or positive contextual experience of that for the animals in each scenario. Additionally, qualitative responses also revealed some weighing up of the extent to which

positive factors may outweigh negative, and the importance of this for overall animal well-being. This is arguably reflective of the wider literature on positive welfare which emphasises the welfare relevance of the affective experience of the animal (e.g. enjoyment, pleasure) and the importance of quality of life (i.e. positive experiences outweigh negative) [44]. Thus findings of this study suggest MOP judgements can be multi-dimensional [1, 50] taking into consideration both the specific contextual factors affecting welfare and how they may contribute to the animal's overall experience of life.

Our results also shed light on when, or for what aspects of animal welfare, MOP consider health and behaviour provision most relevant. Health provision was considered most important for the physical health of the animal and overall well-being. Such judgements reflect the view of animal science that minimising health issues supports the physical and biological functioning of farm animals [51, 52], whilst also being baseline important for general well-being [53]. Conversely, natural behaviours were considered most important for the mental health of farm animals. This closely aligns with scientific perspectives that being able to engage in natural behaviours (e.g. rooting, grazing, nesting, play) promotes positive mental and affective states [e.g. 42, 54]. Such findings further point to the nuanced or multi-dimensional nature of MOP assessments of animal welfare; not only do they alter according to the conditions an animal is in but also according to the aspect of animal welfare in question. Our findings suggest that MOP may consider minimising health issues most relevant to physical health and overall well-being, and promoting natural behaviours most relevant for supporting mental health.

MOP characteristics (e.g. socio-demographic variables) were found to play a minor role, with only a small number of the included characteristic variables having a significant impact on attitudes to health and natural behaviours and welfare-related judgements. Nevertheless, the nature of their impact is reflective of the wider literature. Compared to participants who regularly eat meat, those who do not (regularly) eat meat (i.e. vegetarians, vegans, flexitarians) judged welfare (i.e. well-being, physical and mental health, productivity) to be lower and were more likely to consider natural behaviours more important for animal well-being. This appears to be in line with research finding animal welfare concerns are one of the primary motivations for adopting a vegan or vegetarian diet [55]. Reflecting research which finds an association with higher BAM and greater empathy towards animals and concern for animal welfare [31, 32], participants with a higher BAM judged well-being and mental health lower in the vignette scenarios, and also felt more strongly (as indicated by higher ratings) that minimising health issues and promoting natural behaviours were important for an animal's overall well-being. Females were also found to place greater emphasis on the importance of natural behaviours for animal well-being than males, clearly reflecting the general conclusion in the literature that females tend to be more 'welfare conscious' [39] and more actively involved in welfare issues [31, 56]. Participants' experience of farming was also important for attitudes to natural behaviours, with individuals who came from a farming background (i.e. grew up on a farm) considering natural behaviours to be less important for overall animal well-being. This could, in part, be due to their experience of and assimilation within the farming community, where animal health is central to farmers' conceptions of welfare [13, 57] and often expressed as their priority [37]. Ethnicity also had a minor impact on attitudes, with participants of Asian ethnicity tending to place less importance on minimising health issues for animal well-being. However, there is a critical lack of consideration of the impact of race, class and ethnicity on attitudes to animal welfare within the literature, with most studies only considering a small number of basic socio-demographic determinants such as gender, income and geographic location [58]. Consequently, it is difficult to theorise why these differences between Asian and participants of other ethnicities were observed.

Interestingly, the effect of participant characteristics was most notable in explaining variances in attitudes to natural behaviours; attitudes to the importance of health appeared more stable with little variance between different socio-demographic factors. Certain personal characteristics (e.g. dietary preferences, BAM) may thus be important determinants of natural behaviour expression being considered an important aspect of welfare to promote. This may be because people with a pre-existing interest in animal welfare (e.g. vegans, vegetarians, females) are more concerned by levels of welfare provision and, hence, consider natural behaviours important [6]. Considering this, and the greater variance in attitudes to natural behaviours between participants, it could be argued that encouraging, for example, higher BAM amongst MOP could give rise to higher expectations for animal welfare within society which in turn may impact regulation and policy in relation to animal welfare standards [5, 59].

However, despite these variances between participant characteristics, the information participants received on the level of health or behaviour provision in the vignette scenarios were the primary determinants of their judgements of welfare attributes. This suggests that when faced with actual real-world decisions, as opposed to reporting attitudes and opinions, the specifics of a situation—in terms of health and behaviour provision—has more of an impact on assessments of farm animal welfare than socio-demographic or individual characteristics. Specifically, lower levels of health and behaviour provision caused participants to judge welfare attributes to be lower, while higher levels of health and behaviour provision caused participants to judge welfare attributes to be higher. Such findings somewhat reflect conclusions in the attitudes-to-animals literature which finds that "while personality traits and belief factors are logically related to attitudes to animals, they rarely account for more than one tenth of the variance" [60, p.145]. As such, it is important to account for how contextual factors (e.g. information provided and how it was conveyed) impact MOP attitudes and judgements of welfare [61].

The generalisability of this study's findings to the UK population are also of contemporary importance in the context of the UK's recent exit from the EU (i.e. Brexit). The EU's recent 'Farm to Fork' strategy—which sets outs the EU's future goals in respect of food policy—indicated an intention to achieve higher welfare standards, partly in response to a recognition that "citizens want this" [62, p. 8]. It is still unclear how Brexit will impact animal welfare standards in the UK but concerns have been raised that it will negatively affect animal welfare and place the UK at a weaker 'mid-point' between the progressive welfare standards of the EU and the lower standards of other nations in international trade [63]. By providing an insight into the factors impacting UK MOP's judgements of welfare and their expectations in relation to it, the findings of this study are relevant for informing UK agri-food policy post-Brexit. In particular, they highlight the levels of importance MOP give to health and natural behaviour, potentially demarking some critical 'red lines' government policy may need to be cautious of crossing in post-Brexit policy development [4]. Overall, the findings of this study are compelling, not least for suggesting that health issues may have more of an impact on MOP's judgements of animal welfare than previously thought and the generalisability of this finding to the UK population (according to gender, age and ethnicity). However, the strength of this finding can of course be questioned based on the subjective nature of the vignettes used to present health issues (not) minimised and natural behaviours (not) minimised. The vignette descriptions were based on how livestock farmers who participated in a prior qualitative interview study (see [36, 37]) described their approaches to managing health and promoting natural behaviours. As such, the aspects of health and behaviours selected for inclusion, such as minimising stress, lameness, freedom of movement and social interaction do not comprehensively capture the multifaceted nature of these factors within animal welfare science. Rather, they represent aspects which farmers deem important. Despite the limitations of this, this approach was taken

to ensure the vignettes more closely reflected real-world conditions whilst also responding to calls for greater use of 'folk-conceptions' of welfare in light of the general lack of consensus within animal welfare science on what is relevant for inclusion within the animal welfare paradigm (see [16]). In addition, it could be argued that a full factorial design, whereby participants receive all four vignette scenarios, would strengthen the study design. However, the presentation of numerous vignettes can result in participant fatigue and increase drop-out rates [64], whilst also potentially violating the assumption of independence. As such, participants were presented with only one vignette to reduce potential fatigue, multicollinearity issues and noise (when presented with all four vignettes it is possible that participants' judgements will be based not only on the information provided in the vignette but also how that compares with other vignettes). Although there are limitations to such a design, by conducting analysis at the population-level (i.e. whole sample) it was possible to determine differences in responses to the vignettes. Thus, the findings of this study are not undermined by its experimental design.

## 5. Conclusion

There is a general consensus that MOP prioritise natural behaviours above other aspects of farm animal welfare. Consequently, many relevant animal welfare stakeholders, such as retailers, assurance schemes and policy-makers have sought to meet these public preferences. The findings of this study have important implications for how we view and model animal welfare and how the perspectives of MOP inform the actions of animal welfare stakeholders. Contrary to predominant views, what MOP want for animal welfare may be much more holistic or multi-dimensional, with their priorities varying according to the context and the element of an animal's welfare (e.g. physical or mental health) in question. Importantly, MOP considered welfare to be at its best when both health issues and natural behaviours were supported. However, minimising health issues had the greatest impact on their judgements overall, particularly when there was a trade-off between minimising health and promoting natural behaviours. This has important implications for how public attitudes and expectations relating to animal welfare are understood; there is a need to view public perspectives of welfare as multi-dimensional, influenced by the situation in question, the context of the animal and personal characteristics. Nevertheless, that a representative sample of UK MOP consider animal welfare to be best when both health issues are minimised and natural behaviours promoted is worthy of greater consideration and further investigation.

## Supporting information

**S1 File.**
(DOCX)

**S1 Data.**
(XLSX)

## Author Contributions

**Conceptualization:** Belinda Vigors, Alistair B. Lawrence.

**Formal analysis:** Belinda Vigors, David A. Ewing.

**Funding acquisition:** Alistair B. Lawrence.

**Investigation:** Belinda Vigors.

**Methodology:** Belinda Vigors, Alistair B. Lawrence.

**Validation:** David A. Ewing.

**Visualization:** Belinda Vigors.

**Writing – original draft:** Belinda Vigors.

**Writing – review & editing:** Alistair B. Lawrence.

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
