## [Decision Letter · Decision Letter 0]

21 Jan 2021

PONE-D-20-38465

Happy or Healthy? How members of the public prioritise farm animal health and natural behaviours

PLOS ONE

Dear Dr. Vigors

Thank you for submitting your manuscript to PLOS ONE. After careful consideration, we feel that it has merit but does not fully meet PLOS ONE’s publication criteria as it currently stands. Therefore, we invite you to submit a revised version of the manuscript that addresses the points raised during the review process.

 Both reviewers see merit in the paper,  but have suggested revisions. One concern (reviewer 1) relates to the experimental design applied.  Specifically, each participant was exposed to only one vignette which means that t the trade-off element of the analysis (between health and natural behaviour) cannot be assessed. This represents and important limitation of the research and, although not a fatal flaw, requires discussion as an important limitation of the research.

Reviewer 2 raises the issue of the UK (post- Brexit) policy context, and has suggested that greater consideration of the relevant policy issues in the context of the UK would focus the introduction and improve the discussion section in relation to the translation of results.

We look forward to receiving your revised manuscript.

Kind regards,

Lynn Jayne Frewer, MSc PhD

Academic Editor

PLOS ONE

Additional Editor Comments:

I have now received 2 reviews of your MS submitted to PLOS one. Both reviewers see merit in the paper, but have suggested revisions. One concern (reviewer 1) relates to the experimental design applied. Specifically, each participant was exposed to only one vignette which means that t the trade-off element of the analysis (between health and natural behaviour) cannot be assessed. This represents and important limitation of the research and, although not a fatal flaw, requires discussion as an important limitation of the research.

Reviewer 2 raises the issue of the UK (post- Brexit) policy context, and has suggested that greater consideration of the relevant policy issues in the context of the UK would focus the introduction and improve the discussion section in relation to the translation of results.

Reviewers' comments:

Reviewer's Responses to Questions

**Comments to the Author**

1. Is the manuscript technically sound, and do the data support the conclusions?

Reviewer #1: Partly

Reviewer #2: Yes

2. Has the statistical analysis been performed appropriately and rigorously? 

Reviewer #1: No

Reviewer #2: Yes

3. Have the authors made all data underlying the findings in their manuscript fully available?

Reviewer #1: Yes

Reviewer #2: Yes

4. Is the manuscript presented in an intelligible fashion and written in standard English?

Reviewer #1: Yes

Reviewer #2: Yes

5. Review Comments to the Author

Reviewer #1: This paper has the potential to make a significant contribution to the animal welfare literature. The qualitative analysis at the end of the paper is well done. However, there are issues with the experimental design and the quantitative analysis that need to be addressed (and may not be retrievable) before that part of the analysis could be published. That being said, I think the approach was interesting and potentially very informative. My major concern is with the experimental design. The factorial design approach relies on the selection of different combinations of vignettes to be exposed to different survey respondents. With four vignettes, each respondent could have responded to all four vignettes in randomized order – which would have been a full factorial design. If it was desired that respondents not respond to all four vignettes then the analysis could have been designed so that respondents responded to three or two vignettes – this would still have allowed for respondents to make different tradeoffs among the two important criteria of health or natural behaviour. However with each respondent only exposed to one vignette (line 146) then the tradeoff element of the analysis is not present.

Specific Comments

Abstract - referring to the factorial survey in the abstract is slightly misleading if each respondent only saw one vignette.

Line 54 – the authors suggest the biggest public concern is about non-normal behaviour but the quote also includes painful procedures which are in a separate category – pain and suffering is not directly related to the ‘unnatural’ concern and is not being addressed directly in the research – possibly worth a mention

Line 115 - while I agree that these two concerns are often key - there are other issues - pain, stress etc. which are not addressed in this research. It might be worth mentioning why they are not included or why the two which are addressed are more important again.

Line 128 – the text mentions gender age and ethnicity as the recruiting characteristics for the sample of respondents – I was curious about ethnicity as a characteristic? Is there evidence of that characteristic being important in this context or is it picking up something else like farm familiarity – eating preferences etc? Is there a regional element to the recruiting or education criteria?

Line 146 - the exposure to only 1 vignette is not really an experimental design. If the respondents were exposed to two of the four vignettes - then there would be six combinations of vignettes and selecting two of those in an experimental design would provide some assessments across the population.

Line 162 – asking the questions about the importance of the different attributes (eg. overall well being, productivity) after the vignettes ensures that exposure to the vignettes primes the responses. In fact in Line 163 – the sentence that currently reads – “they would rate (i.e. judge) several attributes relevant to animal welfare” should say “relevant to animal welfare as described for the particular farm they were exposed to”. This is what the researchers wanted – which is ok. However, it is important to identify here and in the discussion of results, that the rankings are not necessarily the same ones that respondents would have identified without the vignettes (or if asked before the vignettes). Given that there are four distinct groups of respondents then the analysis is showing how the vignettes influence respondent responses – although it is then important to know if the four groups have any distinct differences.

Line 175 – since asked after the vignettes, then there is also a priming element to the responses to these questions. That priming element needs to be well described. Again these responses are not separate from the vignettes - have similar questions been asked in other research – without vignettes? If so then the results here by vignette could be compared to other results to illustrate the importance of the vignette priming.

Line 197 - the fact that ethnicity was available ( as a demographic characteristic) does not necessarily result in a need to include in the analysis - the later inclusion of this variable needs to be somehow justified based on previous research or hypothesized relationships. It might be confounded with some other demographic characteristics or even beliefs such as BAM, for example.

Line 197 – the inclusion of the BAM scale after the vignette might also influence results/answers to the BAM questions. Is it possible to compare results of the BAM question responses for the four different vignette groups to other BAM results from other UK studies?

Line 207 - were these tests (normality, correlation etc.) also done by vignette treatment? or just for the overall sample?

Line 231 – need to know if any of the sample demographic and behavioural characteristics, BAM etc. are different by vignette group. If there are differences by vignette group then correlations are possible between the characteristics and the vignette responses.

Line 245 – again if there are demographic or behavioural differences across the vignette groups then the sentiment /content analysis will be affected. This needs to be addressed before the results are described.

Line 294 – need to also show the data by vignette group and compare to UK Census characteristics

Line 300 – the results here are predicated on the vignette exposed to – as described by the authors. However, if the vignette respondent groups differ in other key ways than just the vignette exposed to – such as BAM for example, then it will not be possible to ascribe the differences in results just to the vignettes. Clarifying that is important for the ability to link content of vignette to assessment of importance of a particular animal welfare attribute.

Line 597 – this is just an observation but if ethnicity is the only characteristic of significance and there is no good conceptual reason for it being important then I wonder if it should e included – again is it correlated with vignette possibly?

Reviewer #2: Happy or Healthy? How members of the public prioritise farm animal health and natural behaviours

Authors: Belinda Vigors, David A. Ewing and Alistair B. Lawrence

A really interesting and well written paper on an important and timely subject. The investigation was introduced well, the design of the study was thorough and suitable for the aim and objectives of the research, and the findings discussed in-depth. I am happy to recommend this manuscript for publication after addressing the following minor revisions.

• What is natural/naturalness? Described by the authors as related to “normality” (line 52), but what does this mean? The authors make clear what unnatural and non-normal factors are (e.g., lines 54 – 56) but not the opposite, or what natural factors may be. The authors do acknowledge that naturalness is difficult to define (line 70), and I agree it is, but some text on what this might be/what the existing definitions are would be useful, especially as the study examines how the public value naturalness and the promotion of natural behaviours in farm animals.

• Who are the “key stakeholders”? (Lines 69, 93, etc.). A list of potential, or actual key stakeholders would be useful here.

• The use of vignettes is clearly explained, however the vignette names (high health, low health, high behaviour, low behaviour) is not clearly linked to these explanations until the end of the paragraph (lines 157 – 159). A very minor edit moving these descriptions to earlier in the text would be helpful to the reader.

• Lines 164 – 165 – do you give descriptors for each of the numeric values in the scale, or just the mid-point (5) and the extreme values (0 and 10)? Again, this is a very minor point, but would help the reader understand the survey method.

• Lines 245 – 257 – Examples of sentiment, i.e., positive, negative and neutral language would be useful here. E.g., “Language categorised as positive/negative/neutral sentiment included …….”. Was this done manually, or using computer software?

• A point to bring up in the introduction (Lines 39 onwards) and discussion (Lines 635 onwards) – why survey the views of the UK public specifically, why not a sample of publics from another country? I think a strong case could be made in the introduction and then discussed later as to the importance of understanding public opinion in relation to the UK leaving the EU and the Common Agricultural Policy and therefore changes in the way that agriculture in the UK will be funded, e.g., public money for public goods, and animal health and welfare being described in policy documents as a public good (i.e., the Animal Health and Welfare Pathway). This would highlight how timely and important this piece of research is.

6. PLOS authors have the option to publish the peer review history of their article (what does this mean?). If published, this will include your full peer review and any attached files.

Reviewer #1: No

Reviewer #2: No

---

## [Author Response · Author response to Decision Letter 0]

8 Feb 2021

Dear Editor, 

Please find attached our revised submission to PLOS ONE for the paper “Happy or Healthy? How members of the public prioritise farm animal health and natural behaviours”. Thank you for your consideration of this paper and for the detailed and constructive feedback from both reviewers. 

Below we address each of the reviewers’ recommendations and concerns in detail, through a point-by-point explanation (presented in red). In particular, we address reviewer one’s concerns relating to the experimental design and the exposure of participants to only one vignette. Our response and design has also been cross-checked by our co-author statistician, David Ewing, ensuring the robustness of the design and the findings. There are no significant imbalances between vignette group in terms of demographic factors, and, as we analysed the vignette findings at the population (i.e. sample level) as opposed to individual level, the exposure of each participant to one vignette is appropriate and in line with recommendations of factorial survey design and prior studies published in PLOS ONE. We similarly take on board reviewer two’s suggestion of highlighting the relevance of Brexit to the study context and have made substantial additions, particularly to the discussion section, in light of this. 

I look forward to hearing from you in due course. 

Sincerely, 

Belinda Vigors

Corresponding Author.  

Reviewer 1:

This paper has the potential to make a significant contribution to the animal welfare literature. The qualitative analysis at the end of the paper is well done. However, there are issues with the experimental design and the quantitative analysis that need to be addressed (and may not be retrievable) before that part of the analysis could be published. That being said, I think the approach was interesting and potentially very informative. My major concern is with the experimental design. The factorial design approach relies on the selection of different combinations of vignettes to be exposed to different survey respondents. With four vignettes, each respondent could have responded to all four vignettes in randomized order – which would have been a full factorial design. If it was desired that respondents not respond to all four vignettes then the analysis could have been designed so that respondents responded to three or two vignettes – this would still have allowed for respondents to make different tradeoffs among the two important criteria of health or natural behaviour. However with each respondent only exposed to one vignette (line 146) then the tradeoff element of the analysis is not present.

Thank you for reviewing our manuscript and providing these recommendations. We provide a detailed response to each specific comment below, which also addresses the potential issues you highlight above regarding the experimental design and its impact on the findings. 

Specific Comments

Abstract - referring to the factorial survey in the abstract is slightly misleading if each respondent only saw one vignette.

In consideration of the description of the factorial survey by Auspring and Hinz (2015) https://methods.sagepub.com/book/factorial-survey-experiments/n2.xml as “The core element of FSs is a multidimensional experimental design. Participants judge stimuli, that is, descriptions of hypothetical situations or objects (vignettes). Within these vignettes, the levels of characteristics (dimensions) are systematically varied. A further crucial aspect of FS methods is the random assignment of vignettes to respondents” we feel that it remains appropriate to refer to our study as having a factorial survey design for the following reasons:

I. The vignettes were created by following a 2 x 2 experimental design, created from the 2 factors (health and natural behaviours) and the two levels of those factors (high health/low health and high behaviour/low behaviour)

II. Participants judged hypothetical situations (i.e. they judged vignette scenario descriptions of how a farmer cared for the health and natural behaviours of their animals)

III. Within each vignette, the characteristics of the two key factors (health and natural behaviours) were systematically varied (i.e. as either high health or low health and as either high behaviour or low behaviour) 

IV. The vignettes were randomly assigned to participants. 

Line 54 – the authors suggest the biggest public concern is about non-normal behaviour but the quote also includes painful procedures which are in a separate category – pain and suffering is not directly related to the ‘unnatural’ concern and is not being addressed directly in the research – possibly worth a mention

Thank you for highlighting a potential confusion here. However, we would argue that this sentence remains applicable in the context it is in and as written. Painful procedures here, as indicated by the examples – (e.g. dehorning of dairy calves) etc. refers to procedures that involve doing something which the public consider is not ‘natural’. In other words, the action of removing calves’ horns results in an outcome (i.e. hornless cows) which is not natural. 

Line 115 - while I agree that these two concerns are often key - there are other issues - pain, stress etc. which are not addressed in this research. It might be worth mentioning why they are not included or why the two which are addressed are more important again.

Certainly, a key challenge with this research, and research on social attitudes to animal welfare more generally, is the various interpretations of ‘health’ and ‘natural behaviours’ that exist and the lack of consensus within the animal welfare literature on criteria for each. For many, the issue of ‘pain’ and ‘stress’ come under an overarching category of ‘health’, as they are issues which need to be ‘minimised’ as opposed to the broad category of ‘natural behaviours’ which need to be ‘promoted’; from the perspective of our study, that is the key qualitative difference. Our previous interview research, which as we mention on line 156 (of tracked changes manuscript) informed the wording of our vignettes and how health was described within them, also indicated that factors such as stress and pain are perceived to be part of a broad healthcare envelope by farmers. As such, as can be seen in Table 1, the vignette wording to indicate high health, which was taken directly from the words used by livestock farmers states “I want my animals to be healthy. To me, this means having them stress free, pain free and injury free, whilst also being aware of any health issues that might be arising and dealing with them”. As such, pain and stress are part of health from a farmers’ perspective and thus were included in this study as such. 

Line 128 – the text mentions gender age and ethnicity as the recruiting characteristics for the sample of respondents – I was curious about ethnicity as a characteristic? Is there evidence of that characteristic being important in this context or is it picking up something else like farm familiarity – eating preferences etc? Is there a regional element to the recruiting or education criteria?

The inclusion of ethnicity is related to the sampling criteria used by the company which provided the MOP participant panel: Prolific. In order to be able to provide a sample which is representative of the UK population, they stratify the sample across three demographic factors; age, ethnicity and gender. This is the reason for the inclusion of ethnicity in this study. In addition, prior research (such as Risley-Curtiss, C., Holley, L. C. and Wolf, S. (2006) “The Animal-Human Bond and Ethnic Diversity,” Social Work, 51(3), pp. 257–268. doi: 10.1093/sw/51.3.257) suggests that when examining perspectives relating to animals it is relevant and prudent to also consider and account for ethnicity differences. Given its relevance for ensuring the representative nature of the sample in this study, and recommendations of its relevance in the literature, we feel it is important it remains included in the analysis (please also see further points on this below, in comment relating to Line 197). 

Line 146 - the exposure to only 1 vignette is not really an experimental design. If the respondents were exposed to two of the four vignettes - then there would be six combinations of vignettes and selecting two of those in an experimental design would provide some assessments across the population.

The design of the factorial vignettes, including the randomising so that participants see only one vignette, is an approach that has previously been used by similar studies in the context of animal welfare, including those previously published in PLOS ONE (e.g. Cardoso et al., 2019: https://journals.plos.org/plosone/article?id=10.1371/journal.pone.0205352). In addition, presenting multiple vignettes to participants can violate the assumption of independence (i.e. that there is no relationship between the participants of different groups of the vignette variables), so we chose not to do this in the study design to reduce the potential multicollinearity issues this may cause. We would also suggest that there is particular strength in presenting participants with only one vignette rather than all four. When presented with all four it is likely that participants judgements will be based not only on the information provided in the vignette but also how that compares to the other vignette scenarios, resulting in greater noise in their responses. By presenting participants with just one scenario, we can better examine causation between the vignette scenario and subsequent judgement. In addition, presenting participants will all four vignettes, where each vignette had several judgement questions attached to it, would have resulted in an extremely lengthy survey (with one vignette, participants still took up to 25 minutes to complete it), so using one singular vignette was important to reduce participant fatigue and limit drop-out rates for survey completion.

With regards to the reviewer’s statement in the summary at the top that “with each respondent only exposed to one vignette then the tradeoff element of the analysis is not present” we agree that tradeoffs are not assessed at the individual level and this modelling approach would not be appropriate to make predictions at that level. However, provided the participants in each vignette group are equivalent and the sample is representative of the population (which we believe we have shown in the manuscript and in our response to the reviewer’s comments), our approach is appropriate to determine population-level (i.e. the entire sample) differences in responses to the vignettes.

Line 162 – asking the questions about the importance of the different attributes (eg. overall well being, productivity) after the vignettes ensures that exposure to the vignettes primes the responses. In fact in Line 163 – the sentence that currently reads – “they would rate (i.e. judge) several attributes relevant to animal welfare” should say “relevant to animal welfare as described for the particular farm they were exposed to”. This is what the researchers wanted – which is ok. However, it is important to identify here and in the discussion of results, that the rankings are not necessarily the same ones that respondents would have identified without the vignettes (or if asked before the vignettes). Given that there are four distinct groups of respondents then the analysis is showing how the vignettes influence respondent responses – although it is then important to know if the four groups have any distinct differences.

Yes, you correctly interpret the purpose of a factorial survey using vignettes here. The main strength of this method is that this more closely emulates real-world decision-making, where more than one factor may have to be considered when reaching a decision or making a judgement (as opposed to in a standard survey were participants are normally asked to provide their attitudinal response to a single dimension). It is for this reason that a further benefit of factorial vignettes often cited in the literature is that they can provide insights into causation; that conclusions can be drawn that the information provided in the vignette causes or influences the judgement (e.g. of animal well-being or animal mental health). Thank you for your suggestion regarding line 163 to help clarify this, we have made this addition to line 173 (please see tracked changes). In addition, we checked the demographic variance between the different groups and there was little imbalance between them. That we did not encounter any significant model fitting issues during our analysis of the vignettes also points to there being no issues relating to the different groups having distinctive differences. 

Line 175 – since asked after the vignettes, then there is also a priming element to the responses to these questions. That priming element needs to be well described. Again these responses are not separate from the vignettes - have similar questions been asked in other research – without vignettes? If so then the results here by vignette could be compared to other results to illustrate the importance of the vignette priming.

We recognised that the vignette that the participant was assigned to may have influenced their attitudinal response to the section on “overall attitudes to the importance of health and natural behaviours”. It is for this reason, as described in line 284, we included the vignette they were previously exposed to as a predictor variable in our analysis of this data. By doing so, we were then able to account for and examine the potential for the information they had read in the vignette influencing their response to questions such as ‘how important is minimising health issues for farm animal well-being’ (i.e. we accounted for potential priming effects in the analysis). To our knowledge, similar questions that specifically focus on the question of minimising health issues and promoting natural behaviours have not been asked in extant research. It would be difficult to make satisfactory comparisons. 

Line 197 - the fact that ethnicity was available ( as a demographic characteristic) does not necessarily result in a need to include in the analysis - the later inclusion of this variable needs to be somehow justified based on previous research or hypothesized relationships. It might be confounded with some other demographic characteristics or even beliefs such as BAM, for example.

As mentioned above, the inclusion of ethnicity is important due to this being a key criteria for how the sample was deemed to be representative of the UK population. In addition, as we discuss in line 775 onwards, ethnicity is a variable which is often overlooked or not included in studies on societal perspectives of animal welfare. As such, we consider it is relevant and important to keep it within the study. With regards to potential confounding effects, as detailed in our data checking and preparation section (2.2.), multicollinearity issues, checked using VIFs, were not encountered for ethnicity. As such, there is no evidence that it may be confounded. 

Line 197 – the inclusion of the BAM scale after the vignette might also influence results/answers to the BAM questions. Is it possible to compare results of the BAM question responses for the four different vignette groups to other BAM results from other UK studies?

Thank you for this suggestion. To the best of our knowledge of other studies which use BAM with participants from the UK we conclude that providing an effective comparison would not be possible because i) this study specifically uses the term ‘farm animal’ in the BAM questionnaire whereas others predominantly use the term ‘animal’ and ii) existing studies use convenience samples with much different characteristics (e.g. a pre-existing interest in animal welfare) to the representative sample of this study. As such, we feel a particular strength of this paper is the insight it provides into UK public attitudes of BAM in relation to farm animals – prior to this study, a representative insight into UK attitudes to BAM was unknown.

Line 207 - were these tests (normality, correlation etc.) also done by vignette treatment? or just for the overall sample?

Model checking was performed by examining diagnostic plots both for the overall sample and at the level of the vignette treatment. We have made an addition to line 219 to make this clearer. 

Line 231 – need to know if any of the sample demographic and behavioural characteristics, BAM etc. are different by vignette group. If there are differences by vignette group then correlations are possible between the characteristics and the vignette responses.

The demographic and behavioural characteristics were checked for balance between the groups and there was no evidence of differences. We have added a line to the data preparation section to make this clearer (line 234). 

Line 245 – again if there are demographic or behavioural differences across the vignette groups then the sentiment /content analysis will be affected. This needs to be addressed before the results are described.

As noted above, the demographic and behavioural characteristics were checked for balance between the vignette groups and there was no evidence of extreme difference. The fairly large sample size of 810 participants and random assignment of the vignettes was effective and did not result in e.g. all members of one ethnicity in one vignette group. 

Line 294 – need to also show the data by vignette group and compare to UK Census characteristics

As the randomisation of the data by vignette group did not result in imbalances between groups in terms of demographic characteristics and given that the only demographic data which could be matched to UK census data is gender, age and ethnicity, we feel that doing this would add additional length and a level of detail which is unnecessary for clearly conveying the overall, key findings of the study. 

Line 300 – the results here are predicated on the vignette exposed to – as described by the authors. However, if the vignette respondent groups differ in other key ways than just the vignette exposed to – such as BAM for example, then it will not be possible to ascribe the differences in results just to the vignettes. Clarifying that is important for the ability to link content of vignette to assessment of importance of a particular animal welfare attribute.

This section (3.2.1: Judgement of welfare attributes between scenarios) provides the initial analysis of the vignette scenario data. It’s primary purpose is to provide the reader with an overarching, descriptive insight into differences between vignette scenarios in terms of how various welfare attributes (e.g. well-being) were judged. In the analysis which follows directly on from this section, detailed in section 3.2.2. we deal with the full impact of both vignette factors and participant characteristics on judgements. As such, the potential concerns raised here are dealt with by the analysis in section 3.2.2. 

Line 597 – this is just an observation but if ethnicity is the only characteristic of significance and there is no good conceptual reason for it being important then I wonder if it should be included – again is it correlated with vignette possibly?

As mentioned above, for reasons relating to how the representative sample was stratified by Prolific, we feel it remains important to include ethnicity in the analysis. In addition, the analysis referred to here (now line 617) is the section which is separate from the vignette scenario section. Here, all participants were given the same questions where they were asked to rate how important they considered health and natural behaviours to be for animal well-being. As such, differences in characteristics between vignette group assigned to is not relevant to analysis of this section nor the finding that participants of different ethnicity were more likely to give a lower rating for the importance of minimising health issues. In addition, as detailed in the analysis in this section (line 617), ethnicity was not the only demographic factor of significance; BAM was also a significant predictor of ratings of the importance of minimising health issues. 

 

Reviewer 2:

A really interesting and well written paper on an important and timely subject. The investigation was introduced well, the design of the study was thorough and suitable for the aim and objectives of the research, and the findings discussed in-depth. I am happy to recommend this manuscript for publication after addressing the following minor revisions.

Thank you for your positive feedback on our manuscript and for your recommendations, which we address below. 

• What is natural/naturalness? Described by the authors as related to “normality” (line 52), but what does this mean? The authors make clear what unnatural and non-normal factors are (e.g., lines 54 – 56) but not the opposite, or what natural factors may be. The authors do acknowledge that naturalness is difficult to define (line 70), and I agree it is, but some text on what this might be/what the existing definitions are would be useful, especially as the study examines how the public value naturalness and the promotion of natural behaviours in farm animals.

Yes, naturalness is quite challenging to define as there is a lack of consensus in the literature on a definition for natural behaviours. We have made an addition to line 51 to provide a broad indication to what ‘naturalness’ is taken to mean in the context of research on public attitudes to welfare. In the survey itself, we also provided a line to describe what the terms ‘health’ and ‘natural behaviours’ could be taken to mean: “Health can be taken to mean the absence or minimisation of illness, disease and physical discomfort. Natural behavioural expression refers to behaviours that animals are internally motivated to exhibit and behaviours that are pleasurable”. We have made an addition to the manuscript (line 166 in tracked changes manuscript) to make this evident. 

• Who are the “key stakeholders”? (Lines 69, 93, etc.). A list of potential, or actual key stakeholders would be useful here.

Thank you for highlighting this, we have included examples were relevant. Please see tracked changes and e.g. lines 70, 96. 

• The use of vignettes is clearly explained, however the vignette names (high health, low health, high behaviour, low behaviour) is not clearly linked to these explanations until the end of the paragraph (lines 157 – 159). A very minor edit moving these descriptions to earlier in the text would be helpful to the reader.

Thank you for highlighting the lack of readability here. We have moved this line up to lines 152 onwards. Please see tracked changes manuscript. 

• Lines 164 – 165 – do you give descriptors for each of the numeric values in the scale, or just the mid-point (5) and the extreme values (0 and 10)? Again, this is a very minor point, but would help the reader understand the survey method.

Only descriptors were given to the extreme values and mid-points. We have made an addition to line 176 to make this clearer. 

• Lines 245 – 257 – Examples of sentiment, i.e., positive, negative and neutral language would be useful here. E.g., “Language categorised as positive/negative/neutral sentiment included …….”. Was this done manually, or using computer software?

In light of this recommendation, we have made an addition to line 262 to make this clearer, specifying that computer software was used to do this and this was then checked manually, and to include some example of what was deemed positive or negative (line 265 onwards). 

• A point to bring up in the introduction (Lines 39 onwards) and discussion (Lines 635 onwards) – why survey the views of the UK public specifically, why not a sample of publics from another country? I think a strong case could be made in the introduction and then discussed later as to the importance of understanding public opinion in relation to the UK leaving the EU and the Common Agricultural Policy and therefore changes in the way that agriculture in the UK will be funded, e.g., public money for public goods, and animal health and welfare being described in policy documents as a public good (i.e., the Animal Health and Welfare Pathway). This would highlight how timely and important this piece of research is.

Thank you for this recommendation. We have made some substantial additions to address this, primarily in the discussion section from lines 808 onwards and in the introduction on line 115, to make the timely policy implications of this study clearer.

---

## [Editor Report · Decision Letter 1]

10 Feb 2021

PONE-D-20-38465R1

Happy or Healthy? How members of the public prioritise farm animal health and natural behaviours

PLOS ONE

Dear Dr. Vigors

Thank you for submitting your manuscript to PLOS ONE. After careful consideration, we feel that it has merit but does not fully meet PLOS ONE’s publication criteria as it currently stands. Therefore, we invite you to submit a revised version of the manuscript that addresses the points raised during the review process.

Thank you for revising the MS and responding to the reviewers’ comments. While responses to Reviewer 2, and associated revisions, have been adequately addressed in the revised MS and /or letter to the editor, there are concerns raised by reviewer about the factorial design which have not been adequately addressed in the revised MS. I would therefore recommend that a further (minor) revision of the MS be developed in the discussion section which addresses the issues raised by Reviewer 1 in relation to potential limitations and advantages of the factorial design used, (and which have been well articulated in the response to reviewers), but may also represent potential limitations of the study design. The authors responses to Reviewer 1 regarding participant sampling (in particular to BAM participants) are, however, appropriately presented in the letter to the editor, and in relation to nationally representative sampling across the UK population.

A rebuttal letter that responds to each point raised above .A marked-up copy of your manuscript that highlights changes made to the original version. You should upload this as a separate file labeled 'Revised Manuscript with Track Changes'.An unmarked version of your revised paper without tracked changes. You should upload this as a separate file labeled 'Manuscript'.

We look forward to receiving your revised manuscript.

Kind regards,

Lynn Jayne Frewer, MSc PhD

Academic Editor

PLOS ONE

---

## [Author Response · Author response to Decision Letter 1]

11 Feb 2021

Dear Editor, 

Please find attached our revised submission to PLOS ONE for the paper “Happy or Healthy? How members of the public prioritise farm animal health and natural behaviours”. Thank you for your consideration of this paper and for the positive feedback on our first review. 

In taking on board your recommendations, regarding the need to address the potential limitations of the factorial design highlighted by reviewer 1, we have made an addition to the discussion section of the paper. Here, we outline the reasons behind the randomising of the vignettes between participants and address how this was dealt with in the analysis. This addition can be seen in the tracked changes manuscript from line 835 onwards.

I look forward to hearing from you in due course. 

Sincerely, 

Belinda Vigors

Corresponding Author.

---

## [Editor Report · Decision Letter 2]

16 Feb 2021

Happy or Healthy? How members of the public prioritise farm animal health and natural behaviours

PONE-D-20-38465R2

Dear Dr. Vigors

We’re pleased to inform you that your manuscript has been judged scientifically suitable for publication and will be formally accepted for publication once it meets all outstanding technical requirements.

Kind regards,

Lynn Jayne Frewer, MSc PhD

Academic Editor

PLOS ONE

---

## [Editor Report · Acceptance letter]

22 Feb 2021

PONE-D-20-38465R2 

Happy or Healthy? How members of the public prioritise farm animal health and natural behaviours 

Dear Dr. Vigors:

I'm pleased to inform you that your manuscript has been deemed suitable for publication in PLOS ONE. Congratulations! Your manuscript is now with our production department. 

Kind regards, 

on behalf of

Dr. Lynn Jayne Frewer 

Academic Editor

PLOS ONE